# Dust Storm Event of February 2019 in Central and East Coast of Australia and Evidence of Long-Range Transport to New Zealand and Antarctica

**Hiep Duc Nguyen [1,2,3,*], Matt Riley [1], John Leys [1] and David Salter [1]**

[1] Department of Planning, Industry and Environment, New South Wales, Lidcombe PO Box 29, Australia; matthew.riley@environment.nsw.gov.au (M.R.); john.leys@environment.nsw.gov.au (J.L.); david.salter@environment.nsw.gov.au (D.S.)

[2] Environmental Quality, Atmospheric Science and Climate Change Research Group, Ton Duc Thang University, Ho Chi Minh City 700000, Vietnam

[3] Faculty of Environment and Labour Safety, Ton Duc Thang University, Ho Chi Minh City 700000, Vietnam

[*] Correspondence: hiep.duc@environment.nsw.gov.au

**Abstract:** Between 11 and 15 February 2019, a dust storm originating in Central Australia with persistent westerly and south westerly winds caused high particle concentrations at many sites in the state of New South Wales (NSW); both inland and along the coast. The dust continued to be transported to New Zealand and to Antarctica in the south east. This study uses observed data and the WRF-Chem Weather Research Forecast model based on GOCART-AFWA (Goddard Chemistry Aerosol Radiation and Transport–Air Force and Weather Agency) dust scheme and GOCART aerosol and gas-phase MOZART (Model for Ozone And Related chemical Tracers) chemistry model to study the long-range transport of aerosols for the period 11 to 15 February 2019 across eastern Australia and onto New Zealand and Antarctica. Wildfires also happened in northern NSW at the same time, and their emissions are taken into account in the WRF-Chem model by using the Fire Inventory from NCAR (FINN) as the emission input. Modelling results using the WRF-Chem model show that for the Canterbury region of the South Island of New Zealand, peak concentration of $PM_{10}$ (and $PM_{2.5}$) as measured on 14 February 2019 at 05:00 UTC at the monitoring stations of Geraldine, Ashburton, Timaru and Woolston (Christchurch), and about 2 h later at Rangiora and Kaiapoi, correspond to the prediction of high $PM_{10}$ due to the intrusion of dust to ground level from the transported dust layer above. The Aerosol Optical Depth (AOD) observation data from MODIS 3 km Terra/Aqua and CALIOP LiDAR measurements on board CALIPSO (Cloud-Aerosol LiDAR and Infrared Pathfinder Satellite Observations) satellite also indicate that high-altitude dust ranging from 2 km to 6 km, originating from this dust storm event in Australia, was located above Antarctica. This study suggests that the present dust storms in Australia can transport dust from sources in Central Australia to the Tasman sea, New Zealand and Antarctica.

**Keywords:** dust transport; Australia; Tasman Sea; New Zealand; Antarctica; WRF-chem; CALIPSO; MODIS

## 1. Introduction

Dust aerosols in the atmosphere can affect the weather by acting as condensation nuclei, forming clouds, scattering and absorbing incoming solar radiation, and therefore influencing the climate [1]. Dust storms travel long distances and impact the regions downwind [2]. In Australia, dust storms frequently occur from deserts in central Australia (including Simpson, Gibson, Great Victoria and Sturt deserts) [3], and from agricultural land that has been in prolonged drought in western South Australia,

Queensland, Victoria and New South Wales (NSW) [4]. As they traverse inland cities and those along the eastern seaboard of Australia, they occasionally affect air quality in these population centres [2,5].

Australia is a dry continent with vast desert inland in central Australia. In their study of global distribution of aeolian dust using the Global Ozone Chemistry Aerosol Radiation and Transport (GOCART) model from 1981 to 1996, Ginoux et al. 2004 [6] showed that Australia is the third largest source contributing to global dust emission, after North Africa and Asia. However, in terms of dust emission amount, it is only about 20% of Asia's and 7% of North Africa's annual emission amount.

Approximately 110 million tons of dust emission from Australia each year are transported along 2 main pathways across the continent, the south-east pathway and the north-west pathway, with approximately 30 million tons deposited to the Tasman sea and Pacific Ocean off the east coast of Australia [7]. Bowler 1976 [8] describes the two major dust paths above. These dust paths were most active between 25,000 and 13,000 BP. Although dust activity was active between 6000 and 3000 BP and in the last 2000 years, it has not been at the level of the early Quaternary [8]. Sprigg 1982 [9] identified several wind systems that cause wind erosion and transport dust in multiple directions in south-eastern Australia.

O'Loingsigh et al. 2017 [10], in their study of the sources and pathways of dust during the Australian "Millennium Drought" decade (late 1990s to mid-2010) using trajectory analysis, showed that dust storm events mostly occurred in the austral spring and summer. Dust transport from the centre of the continent to the Australian east coast is typically induced by the cold front systems from the Southern Ocean. In general, the dust is transport by three wind systems, as seen from south to north on the Australian east coast: the Southern Dust Path, below 35° S, due to prefrontal northerlies carrying dust to the south coast of Victoria; the Eastern Dust Path (35° S to 25° S, from Canberra, Sydney to Brisbane) due to the frontal westerlies; and the North East Dust Path (25° S to 20° S) due to the postfrontal south westerlies [10].

Recent notable events that impact cities on the eastern seaboard include: the 22nd October 2002 event [4], the "Red Dawn" dust storm event in September 2009 [2], and undocumented events in November 2018 and February 2019. The geomorphic literature has also documented long-range dust transport in paleo-environments [11].

Many dust storm events have transported and deposited dust in the Tasman sea [12–14], and dust can be transported as far as New Zealand [15–18] and Antarctica [19]. Some records of ice cores obtained from Antarctica have shown dust transport originating from Australia [19,20]. Long-range dust deposition enhances carbon cycling [7], iron fertilisation of the ocean, and the promotion of phytoplankton growth [21,22]. Following the results of their study on the dust pathways on the east coast of Australia, O'Loingsigh et al. 2017 [10] suggested that the phytoplankton response to dust deposition in the Great Barrier Reef off Queensland was driven by dust from the North East Dust Path. Phytoplankton response in the Tasman Sea is due to deposited dust from the Eastern Dust Path, and dust-induced phytoplankton growth in the Southern Ocean is most likely a result of dust from the Southern Dust Path.

Dust particle size decreases as the distance downwind increases, due to larger particles such as quartz and feldspar sand having a greater settling velocity than smaller particles of clay minerals [23]. However, some particles of larger size, such as giant mineral dust particles (>75 μm) which are not usually modelled, have been found further away than expected from their sources due to turbulence and uplift in the convective system [24].

Dust transport studies have used extensive available sets of data from satellites and ground-based observation. Chen et al. 2016 [25] studied the transport of dust events in March 2010 from the Taklimakan and Gobi deserts across China to Korea and Japan using the WRF-Chem model; they used Aerosol Optical Depth (AOD) data from the AERONET ground network and vertical aerosol extinction coefficients from the CALIPSO satellite. Chen et al. 2018 [26] recently studied intercontinental dust transport from two dust storm events in 2015 in North African (Sahara-Sahel) and Asian (Taklamakan) deserts to the East and West coast of the U.S. using AOD from MYD04-3k (Aqua) Moderate-Resolution

Imaging Spectroradiometer (MODIS) level 2 products, AI (Aerosol Index) or Aerosol Optical Thickness (AOT) derived from satellite measurements such as GOES (Geostationary Operational Environmental Satellite system), Aerosols/Smoke Product (GASP), OMI (Ozone Monitoring Instrument) onboard the Earth Observing System (EOS), and VIIRS (Visible Infrared Imaging Radiometer Suite) aboard the Suomi National Polar-orbiting Partnership (Suomi-NPP). In addition, LiDAR measurements of aerosol vertical profile by CALIOP onboard the CALIPSO satellite, ground-based AERONET of solar radiation and air quality monitoring stations, as well as MERRA-2 aerosol reanalysis, were also used to characterise the source and long-range transport (LRT) patterns of dust events from North Africa and Asia to the U.S.

The WRF-Chem model has been used with various dust emission schemes by many authors to study the dispersion of dust in the atmosphere from dust sources such as the Sahara, Arabian and Central Asian deserts [23,25,27–33]. There are 5 dust emission schemes available in WRF-Chem: GOCART (Georgia Tech/Goddard Global Ozone Chemistry Aerosol Radiation and Transport), AFWA (Air Force Weather Agency), and three of the UoC (University of Cologne) schemes (Shao 2001, Shao 2004 and Shao 2011) [34–36].

Comparisons of these dust emission schemes in WRF-Chem have been conducted recently by many authors, such as [29,31,33,37,38] in their simulation studies of dust emission and dispersion. The prediction from different schemes was compared with observation data such as MODIS AOD. Compared to the simpler GOCART algorithm, the AFWA and Shao schemes generally performed better due to their inclusion of saltation mechanisms in generating dust from wind speed. Yuan et al. 2019 [33] suggested that the differences in their results from using different schemes (GOCART, AFWA and Shao 2004) were dependent on the sensitivities of the threshold friction velocity on the surface properties of the dust source.

Ma et al. 2019 [39] evaluated dust schemes in four air quality models—CMAQS v.5.2.1, CAMx v6.50, CHIMERE v2017r4, and WRF-Chem v3.9.1—in a simulation study of the springtime 4–6 of May 2015 dust storm in East Asia. Three dust schemes in WRF-Chem, two schemes in CMAQS and CHIMERE, and one in CAMx were used in the simulation. They found that WRF-Chem with the AFWA dust scheme using the seasonal dust source map from Ginoux et al. 2012 [40] and parameter tuning showed the best performance when compared with observation from ground and satellites (MODIS AOD, CALIPSO) followed by WRF-Chem using the UoC_Shao2011 dust scheme.

In this study, the dust event of 12–15 February 2019 is studied with a focus on the long-range transport (LRT) patterns of dust across the eastern seaboard of Australia, the Tasman Sea, and beyond to Antarctica using the WRF-Chem model with GOCART/MOZART aerosol chemistry, as well as observed MODIS satellite AOD data and aerosol vertical profile from CALIOP LiDAR. The AOD data from satellites, air quality monitoring data at various stations in Australia and New Zealand, MERRA-2 reanalysis, and LiDAR vertical profile of aerosols from CALIOP measurements on board CALIPSO satellite and MODIS AOD data are used for comparisons with modelling results. The CALIOP LiDAR data is particularly valuable, especially in remote places where there are no ground measurements, and the vertical structure of aerosols distribution is required in order to understand the tropospheric and stratospheric transport of aerosols.

## 2. Data and Methods

The data used in this study consists of MODIS Aqua/Terra satellite AOD data, CALIOP LiDAR vertical structure of aerosols, Fire Inventory from NCAR (FINN), air quality monitoring data from the Department of Planning, Industry and Environment (DPIE) air quality monitoring network in New South Wales, and the Canterbury Regional air quality monitoring stations in New Zealand. The National Centre for Environmental Prediction (NCEP) Final Analysis (FNL) Reanalysis data provides the boundary and initial meteorological condition to be used in the WRF-Chem meteorological and chemistry model.

The MODIS 3 km aerosol products MYD04-3k (Aqua) and MOD04-3k (Terra) and MOD08_D3 (Terra), MYD08_E3 (Aqua) products were retrieved from Level-1 and Atmosphere Archive & Distribution System (LAADS) https://ladsweb.modaps.eosdis.nasa.gov. The MODIS MOD08_D3, MYD08_E3 is a level-3 MODIS gridded atmosphere daily global joint product providing $1° \times 1°$ grid average values, while the MYD04-3k and MOD04-3k level 2 products monitor the ambient aerosol optical properties (e.g., optical thickness and size distribution), mass concentration, look-up-table-derived reflected and transmitted fluxes, as well as quality assurance and other ancillary parameters, over the oceans globally and over a portion of the continents. Both are suitable for AOD verification with model prediction. The MOD08_D3 product, however, is the most suitable, as it combines both Dark Target (DT) algorithm over ocean and part of the continents and Deep Blue (DB) algorithm over bright surfaces such as desert.

## 2.1. Dust Emission Model

The above data are used in conjunction with dust modelling using WRF-Chem to corroborate and verify the model performance. There are several dust scheme options used in WRF-Chem; namely, the GOCART (Georgia Tech/Goddard Global Ozone Chemistry Aerosol Radiation and Transport) aerosol scheme, the GOCART with AFWA (Air Force Weather Agency) modification to include saltation flux and the University of Cologne (UoC) based on three different dust emission algorithms (Shao 2001, Shao 2004 and Shao 2011) [34–36].

The original GOCART model for calculating dust emissions for different particle sizes based on wind speed and soil moisture was described by Ginoux et al. 2001 [41]. The empirical equation for calculating dust flux for each particle bin size p as related to wind speed is given as

$$F_p = \begin{cases} CSs_p\, U^2(U - U_t(D_p,\, \theta_s)) & U > U_t(D_p,\, \theta_s) \\ 0 & U \le U_t(D_p,\, \theta_s) \end{cases} \tag{1}$$

where C is an empirical constant ($10^{-6}$ g s$^2$ m$^{-5}$), S is a unit dust source strength function or erodibility indicating the availability of particles to be entrained, $s_p$ is the mass fraction of emitting dust from soil class (sand, silt and clay) of size p, U is the wind speed at 10 m and $U_t$ ($D_p$, $\theta_s$) is the threshold wind velocity over which erosion occurs, $D_p$ is the particle diameter of size p and $\theta_s$ is the degree of saturation as measurement of soil moisture.

The C constant, as implemented in the WRF-Chem source code, is $0.8 \times 10^{-6}$ g s$^2$ m$^{-5}$. Eltahan et al. 2018 [31] used this as a tuneable value to closely match the predicted AOD with MODIS AOD measurements. Chen et al. 2018 [26] used a value of 1.2 μg s$^2$ m$^{-5}$ to match the MODIS ADO. In addition, Ginoux et al. 2012 [40] used C as $10^{-6}$ g s$^2$m$^{-5}$ for topographic erodibility and C = $1.9 \times 10^{-6}$ g s$^2$ m$^{-5}$ for soil erodibility derived from MODIS Deep Blue Level 2 satellite product.

The unitless source strength constant S as a measure soil erodibility is obtained from the map representing the fraction of cell area of erodibility of bare soil of sand and clay. In the WRF-Chem model, the erodibility field is generated in WRF Pre-processor (WPS) using the topographic data before being used in the WRF-chem core. S is calculated based on the premise that the dust material available at the low point is the result of alluvial processes [42]:

$$S = \left( \frac{z_{max} - z_i}{z_{max} - z_{min}} \right) \tag{2}$$

where $z_i$ is the elevation of cell I, and $z_{max}$, $z_{min}$ are the maximum and minimum elevations in the surrounding $10^0 \times 10^0$ area, respectively. This topographically based dust source function (DSF) above was referred to as the Ginoux source function by [43].

We acknowledge that this dust source domain underestimates the area that actually contributes to the dust plume. This is because during droughts, some parts of the landscape that are not topographically low, but have reduced ground cover, become dust sources. These are not accounted for

in the WRF Pre-processor (WPS). The monthly DustWatch report for February 2019 [44] describes the dust sources in the agricultural land of western New South Wales. Figure 1 shows the total vegetation cover for Australia in February, June, November 2018 and February 2019, as determined by the method of Guerschman et al. 2015 [45]. Values less than 50% indicate possible source areas. It can be seen from these figures that the drought affected south east Queensland and most of western New South Wales from late 2018 and early 2019, when the dust storm occurred on 11–15 February 2019. Extra dust sources from drought-affected areas include part of western NSW near the border with the states of South Australia and Victoria.

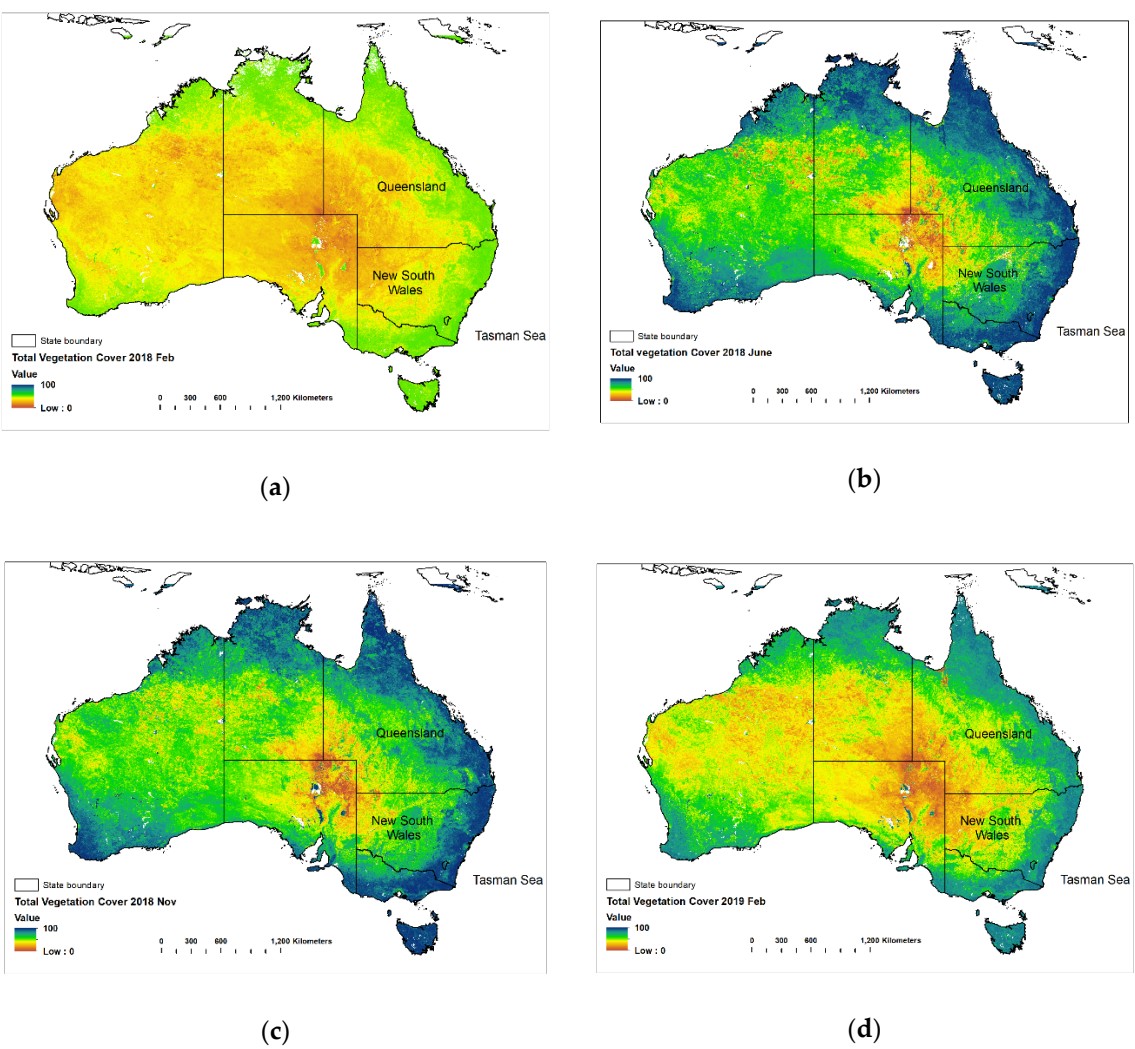

**Figure 1.** Total vegetation cover (%) for Australia in February 2018 (**a**), June 2018 (**b**), November 2018 (**c**) and February 2019 (**d**) (source: http://www-data.wron.csiro.au/remotesensing/MODIS/products/public/v310/australia/monthly/cover/).

The simple empirical GOCART scheme is popular, as it provides a relationship between wind speed and soil characteristics with dust flux emission in a simple equation. The AFWA extension of the GOCART scheme and the UoC schemes (Shao 2001, Shao 2004 and Shao 2011) [34–36] include the saltation mechanism of dust generation from the bombardment of dust particles with various degrees of detail. In this study, the GOCART-AFWA scheme is used to study the dust emission and transport from the February 2019 dust storm event. Here we describe the dust emission model of this scheme as implemented in WRF-Chem.

The most important variable in Equation (1) is the threshold wind velocity. As soil erodibility is dependent on soil moisture, the threshold velocity is also adjusted for soil moisture. In the WRF-Chem implementation of GOCART, the threshold velocity is, however, calculated as a threshold friction velocity based on an equation from Marticorena and Bergametti, 1995 [46]. The Marticorena-Bergametti equation effectively replaces the threshold wind speed $U_t$ used in Ginoux flux Equation (1) with threshold friction velocity $u_{*t}$ that was derived in terms of friction velocity $u_*$ rather than wind speed U at 10 m:

$$u_{*t} = 0.129 \frac{\left(\frac{\rho_p g D_p}{\rho_a}\right)^{0.5}\left(1 + \frac{0.006}{\rho_p g D_p^{2.5}}\right)^{0.5}}{\left[1.928\left(a\left(D_p\right)^x + b\right)^{0.092} - 1\right]^{0.5}} \tag{3}$$

where $D_p$ is the diameter of size p particle, g is gravitational acceleration, $\rho_p$ is density of size p particle, $\rho_a$ is air density, x = 1.56, a = 1331 cm$^{-x}$ and b = 0.38.

As Legrand et al. 2019 [42] noted, the Marticorena-Bergametti equation was designed for determining the threshold of initiating wind-shear-based saltation of grain, and not for representing the threshold for the emission of dust particles from the surface, and hence the AFWA scheme uses it to include the saltation bombardment and particle disaggregation mechanism of dust production.

The AFWA scheme uses a correction factor to account for the effect of soil moisture on threshold friction velocity, as follows:

$$u_{*t,s,p} = u_{*t}\left(D_{s,p}\right)f(\theta) \tag{4}$$

where

$$f(\theta) = \begin{cases} \sqrt{1 + 1.21\left(\theta_g - \theta_g'\right)^{0.68}} & \theta_g > \theta_g' \\ 1 & \theta_g < \theta_g' \end{cases} \tag{5}$$

$D_{s,p}$ is the diameter of saltation size bin p, while $\theta_g$ is the gravimetric soil moisture fraction and $\theta_g'$ is the soil moisture fraction to be absorbed into the soil before capillary forces start to influence the particle disaggregation, and as such is a kind of threshold moisture.

The gravimetric soil moisture is provided from the volumetric soil moisture $\theta_v$ by an equation in WRF-Chem:

$$\theta_g = \frac{\rho_w \theta_v}{(2.65 - 0.15c_s)(1 - \phi)} \tag{6}$$

and

$$\theta_g' = 0.0014(100c_s)^2 + 0.17(100C_s) \tag{7}$$

where $\rho_w$ is water density, which is 1 gcm$^{-3}$, $\phi$ is the soil porosity, $c_s$ is the soil clay content mass fraction, and the term $(2.65 - 0.15c_s)$ represents the soil density.

Once the threshold friction velocity has been determined, the horizontal saltation flux is next found in order to find the bulk dust emission vertical flux.

The horizontal saltation flux for particle size of bin *p* is determined as follows:

$$H\left(D_{s,p}\right) = \begin{cases} C_{mb}\frac{\rho_g}{g}u^{*3}\left(1 + \frac{u^*_{t,s,p}}{u^*}\right)\left(1 - \frac{u^*_{t,s,p}^2}{u^*}\right) & u^* > u^*_{t,s,p} \\ 0 & u^* \leq u^*_{t,s,p} \end{cases} \tag{8}$$

The constant $C_{mb}$ is implemented in GOCART AFWA WRF-Chem model with a value of 1 based on Marticorena and Bergametti 1995 [46], rather than the value of 2.61 presented in the original equation. Eltahan et al. 2018 [31] used $C_{mb}$ as a tuneable parameter.

In this study, this parameter was also used a tuneable constant, which could be adjusted in order to be able to compare the predicted PM$_{10}$ ground dust concentration with the PM$_{10}$ observed at monitoring stations. Eltahan et al. 2018 [31] adjusted the value of this parameter within a range from 1 to 10 in their studies of two dust storm events in January 2004 and March 2013 in the deserts of Egypt.

As noted in the WRF-Chem code, the sandblasting efficiency β acts to reduce the dust emissions in areas with a few dust-sized particles so that there is a decrease in lofting efficiency; otherwise, super sandy zones would be huge dust producers [46]. The value of β varies between $1 \times 10^{-6}$ and $1.36 \times 10^{-6}$ cm$^{-1}$ (varying with clay mass fraction from 0 to 1).

Finally, the dust emission flux is calculated for regions defined as grassland, sparsely vegetated, or barren [42], i.e., regions that have a surface roughness length $z_0 \leq 20$ cm, as shown in Equation (9):

$$F_B = \begin{cases} GS\beta & z_0 \leq 20 \text{ cm} \\ 0 & z_0 \geq 20 \text{ cm} \end{cases} \tag{9}$$

The dust emission models above used a number of parametrisation equations with some defined constant variables, such as threshold friction velocity in the AFWA and UoC schemes, which can affect dust emissions, as well as the results of dust concentration and the particle transport in different regions and in different dust storm simulation cases, depending on the characteristic values of those constants. Tuning scales or coefficients have been used in dust schemes such as GOCART to calibrate the scheme to account for uncertainties in soil characteristics of a particular area and to obtain results that can be compared with observation data such as those from AOD [27,31,38,47].

In WRF-Chem, the GOCART-AFWA scheme is implemented with volumetric soil moisture, which can be tuned with a dust_smtune option in namelist, such that $\theta_v$ in Equation (6) has the value $\theta_v = \max(\theta_v{*}\text{smtune}, 0)$.

### 2.2. Fire Emission Using FINN and WRF-Chem MOZART Gas-Phase Chemistry

In the New England area of northern NSW, there were large and extensive wildfires during the period 10–15 February 2019. The emissions from the wildfires affected the air quality in this region and beyond, as shown by the MODIS Terra/Aqua satellites. To account for this source of emission of both aerosols and gaseous chemical species, Fire Inventory from NCAR (FINN) emission data with a resolution of 1 km, derived from MODIS Rapid Response fire count (FIRMS) hotspots, are used. These data are usually provided on a daily basis and can be downloaded from https://www.acom.ucar.edu/acresp/dc3/finn-data.shtml. We use FINN instead of the Global Fire Emissions Database, Version 3 (GFED v3), as GFED v3 only provides monthly temporal resolution at 0.5° spatial resolution, as opposed to the daily resolution of FINN.

The FINN emission data are prepared with the appropriate species for 3 gas-phase chemical mechanisms as provided by FINN: GEOS, SAPRC99 and MOZART chemistry. The files provided by FINN are named as GLOB_GEOSchem_yyyyJ.txt.gz, GLOB_MOZ4_yyyyJ.txt.gz or GLOB_SAPRC99_yyyyJ.txt.gz where yyyy is the year y and J is the Julian day of the year.

In this study, we use the MOZART gas-phase chemistry model, as it can be used in conjunction with the GOCART aerosol chemistry module in WRF-Chem. The WRF-Chem chemistry option is then MOZART/GOCART (chemp_opt = 112).

### 2.3. WRF-Chem Configuration

The WRF-Chem domain for this study consists of the eastern half of Australia, the Tasman sea, and New Zealand in order to adequately cover the region of interests for the dust storm event of February 2019. The Reanalysis data from the National Centre for Environmental Prediction (NCEP) Final Analysis (FNL) provides the boundary and initial meteorological conditions, as well as the default chemical profiles in WRF-Chem for the simulation. Figure 2 shows the domain configuration of 355 east–west grid points and 218 north–south grids of 15 km by 15 km resolution and 30 vertical levels.

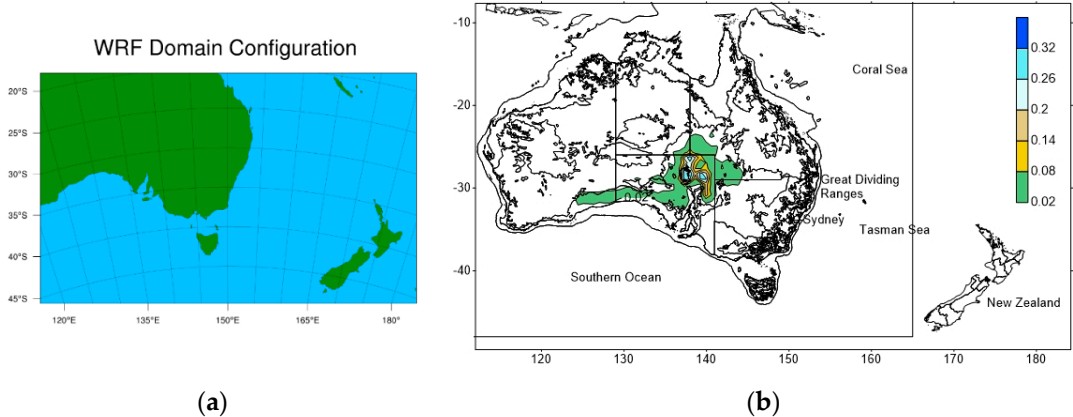

**Figure 2.** (**a**) Modelling domain (Lambert projection) with 15 km × 15 km resolution (lat from −45.5384° to −17.914°, and longitude from 116.025° to 184.175°). (**b**) The topographic sand erodibility (0–1) contour map (coloured) within the topography of the Australian continent is used by the WRF-Chem dust module to scale the dust flux.

To obtain outputs for the aerosol optical properties (extinction coefficient, extcof55, at 550 nm wavelength), we chose the aerosol chemistry GOCART (chem_opt = 300) and the output of the aerosol extinction coefficient calculated based on the Maxwell-Garnett approximation (aer_opt_opt = 2). As the interactive feedback of radiative aerosols in meteorology can affect dust emissions, as shown by Wang et al. 2018 [48], and similarly by Rizza et al. 2017 [30], who considered the aerosol chemistry GOCART scheme in their study of a dust outbreak from the Sahara Desert in 2004 as well as the Maxwell-Garnett rule for deriving aerosol properties, this WRF-Chem simulation study of the dust event on the east coast of Australia also considers the direct radiative effect of aerosol using Rapid Radiative Transfer Model (RRTMG) for both short-wave and long-wave radiation (ra_sw_physics = 4, ra_lw_physics = 4). The direct radiative effect of aerosol on meteorology determined by RRTMG was also used by Zhao et al. 2011 [49] and Chen et al. 2014 [28] in their studies of the effect of aerosols on precipitation in West Africa and dust mass balance from Asian desert dust sources.

The topographic sand erodibility field, as shown in Figure 2b, is generated from the map of sand and clay bare soil and topographic data, as represented by Equation (2). The simulated wind speed and soil moisture in WRF provide the inputs for the WRF-Chem dust emission scheme. The microphysics scheme used is the WRF Single-Moment (WSM) 3-class suitable for mesoscale grid sizes (mp_physics = 3), and the surface layer physics is based on the Monin-Obukhov similarity scheme (sf_sfclay_physics = 1), which is compatible with the planetary boundary layer (PBL) YSU (Yonsei University) model. While the land surface model used is the Noah Land-Surface Model, a unified NCEP/NCAR/AFWA scheme with soil temperature and moisture in four layers, fractional snow cover and frozen soil physics (sf_surface_physics = 2), the cumulus option is the Kain-Fritsch scheme (cu_physics = 1). Table 1 summarises the parametrisation used in the namelist setting of WRF-Chem for this study, along with other relevant dust studies in literature.

**Table 1.** Namelist setting in WRF-Chem setup.

| Physical Parametrisation | Namelist Variable | Option | Model/Scheme | Other Relevant Dust Studies |
|---|---|---|---|---|
| Microphysics | mp_physics | 3 | WRF Single Moment | Chen et al., 2016, Yuan et al., 2019 (two-moment Morrison scheme) |
| Land surface | sf_surface_physics | 2 | Noah Land-Surface Model | Cavazos-Guerra and Todd 2012, Zhao et al., 2010, Eltahan et al., 2018, Yuan et al., 2019 |
| Surface layer physics | sf_sfclay_physics | 1 | Monin-Obukhov similarity | Fountoukis et al., 2016 |
| Planetary Boundary Layer | bl_pbl_physics | 1 | YSU scheme | Fountoukis et al., 2016, Eltahan et al., 2018, Chen et al., 2018. Yuan et al., 2019 |
| Shortwave radiation | ra_sw_physics | 4 | Rapid Radiative Transfer Model (RRTMG) | Rizza et al., 2017, Eltahan et al., 2018, Chen et al., 2018, Yuan et al., 2019 |
| Long wave radiation | ra_lw_physics | 4 | Rapid Radiative Transfer Model (RRTMG) | Rizza et al., 2017, Fountoukis et al., 2016, Eltahan et al., 2018, Chen et al., 2018, Yuan et al., 2019 |
| Aerosol Chemistry | chem_opt | 300 (112) | GOCART (and MOZART/GOCART) | Rizza et al., 2017 |
| Dust scheme | dust_opt | 3 | GOCART-AFWA scheme | Fountoukis et al., 2016, Yuan et al., 2019 |
| Aerosol extinction coefficient approximation | aer_opt_opt | 2 | Maxwell-Garnett approximation | Rizza et al., 2017 |
| Aerosol radiative feedback | aer_ra_feedback | 1 | Turn on aerosol radiative feedback with RRTMG model | Wang et al., 2018, Cavazos-Guerra and Todd 2012 |

The GOCART-AFWA distributes dust particles into 5 bins: bin 1 (particles of size 0–1 μm), bin 2 (1.0–1.8 μm), bin 3 (1.8–3.0 μm), bin 4 (3.0–6.0 μm), and bin 5 (6.0–10 μm). The effective radii of particle sizes in those bins are 0.5, 1.4, 2.4, 4.5, and 8 μm, respectively. Particles with radii of between 63 μm and 2000 μm are considered to be sand; between 2 μm and 63 μm are considered to be silt; and below 2 μm are considered to be clay particles [31,50]. The dust fractions in the 5 dust bins were designated as 0.1024, 0.1012, 0.2078, 0.4817 and 0.1019, respectively [33]. The concentration output is in μg/kg dry-air (which is equal to 1.29 μg/m$^3$ at standard temperature and pressure).

The AOD at 0.55 μm is calculated as

$$AOD = \sum_{i=1}^{n-1} coef55 * \Delta z_i \tag{10}$$

where $\Delta z_i$ is the i-layer depth ($z_i + 1 - z_i$) and *coef55*, as output from WRF-Chem, is the extinction coefficients for 55 μm wavelength.

The biogenic and anthropogenic emission data for WRF-Chem input used in this study are the MEGAN (Model of Emissions of Gases and Aerosols from Nature) and the global 2005 anthropogenic EDGAR (Emission Database for Global Atmospheric Research) version 4 compiled for the Task Force on Hemispheric Transport of Air Pollution (TF-HTAP), which has emission sources including land-based and shipping transport at $1° \times 1°$ resolution. The EDGAR-HTAP emission data includes fine particulate matter $PM_{10}$ and $PM_{2.5}$, carbonaceous speciation Black Carbon (BC) and Organic Carbon (OC), ozone precursor gases such as Carbon Monoxide (CO), Nitrogen Oxides ($NO_x$), Non-Methane Volatile Organic Compounds (NMVOC), acidifying gases such as Ammonia ($NH_3$), Nitrogen oxides ($NO_x$) and Sulphur Dioxide ($SO_2$) and the greenhouse gases $CO_2$, Methane ($CH_4$) and Nitrous Oxides ($N_2O$). Figure 3 shows EDGAR_HTAP anthropogenic emission of NO and BC in the modelling domain.

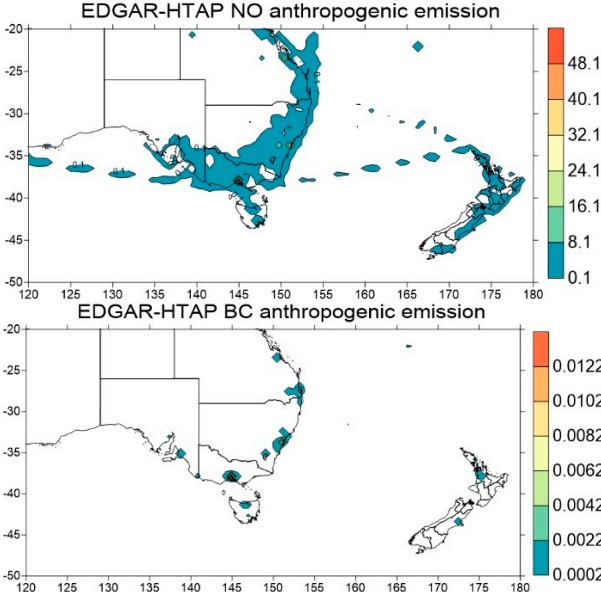

**Figure 3.** EDGAR-HTAP global anthropogenic emission of nitrogen monoxide (NO) emission (mol $km^{-2}$ $h^{-1}$) and black carbon (BC) emission (µg/$m^3$ m/s) for February 2019. The emission includes land-based and ocean shipping emission.

## 3. Results

Figure 4 shows the spatial distribution at level 1 (ground) of dust particle concentration for bin 2 sizes at 13 February 2019 05:00 UTC (13 February 2019 15:00 AEST) and the following day on 14 February 2019 4:00 UTC (14 February 2019 14:00 AEST or 16:00 NZ) as predicted by the WRF-Chem GOCART-AFWA dust model. Peak $PM_{10}$ and $PM_{2.5}$ concentrations were detected at the Tamworth and Armidale monitoring sites in the New England region of NSW on 13 February 2019 6:00 UTC (13 February 2019 16:00 AEST) and Bargo, Bringelly, Prospect and other monitoring stations in the Sydney region, Australia on 12 February 2019 9:00 UTC (12 February 19:00 AEST), and the monitoring sites in the Canterbury region of the South Island in New Zealand on 14 February 2019 5:00 UTC (14 February 2019 17:00 NZ).

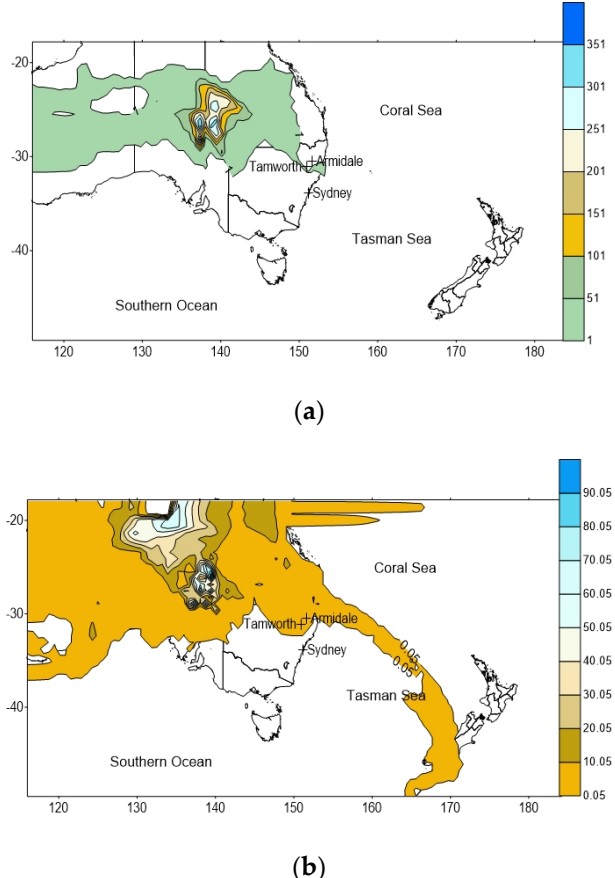

**Figure 4.** Concentration of dust particles (µg/kg dry-air) of bin 2 size (1 to 1.8 µm) on (**a**) 13 February 2019 5:00 UTC and (**b**) 14 February 2019 4:00 UTC.

The predicted temperature and surface wind field from WRF-Chem across the domain show a consistent wind pattern from south westerly on 11–12 February 2019 12:00 UTC to easterly on 13 February 2019 12:00 UTC and 14 February 2019 12:00 UTC, with a cold front in the Southern Ocean, as shown in Figure 5.

The wind fields at different heights and times are also important in understanding the dispersion and transport of dust aerosols. The temperature and wind field at 700, 850 and 925 mb (hPa) levels in the lower troposphere on 11 February 2019 and 12 February 2019 at 12:00 UTC are shown in Figure A1 of the Appendix A. At 925 mb and 850 mb, the south westerly flow from the Southern Ocean merged with the easterly flow from the Coral Sea, forming a northerly and westerly flow above the Tasman Sea and New Zealand. The wind field pattern at 700 mb above the Tasman Sea and New Zealand is similar to those at 925 mb and 850 mb. For the 13 February 2019 12:00 UTC, the predicted wind from WRF-Chem at 925 mb, 850 mb and 700 mb shows south westerly and southerly flow over the Australian continent, and northerly and westerly over the Tasman Sea and New Zealand, while on 14 February 2019 12:00 UTC, the wind fields at 925, 850 and 700 mb show a westerly flow over the Southern Ocean and New Zealand and a southerly flow above the Tasman Sea (see Appendix A Figure A2).

The CALIOP LiDAR, onboard the CALIPSO satellite, detected a thick layer of dust (up to 2 km above ground level) above central Australia on 11 February 2019 from 4:27 to 4:40 UTC, as shown in Figure 6. The extent of the dust cloud ranged from western New South Wales near the border with Victoria to western Queensland, and was approximately 1500 km long.

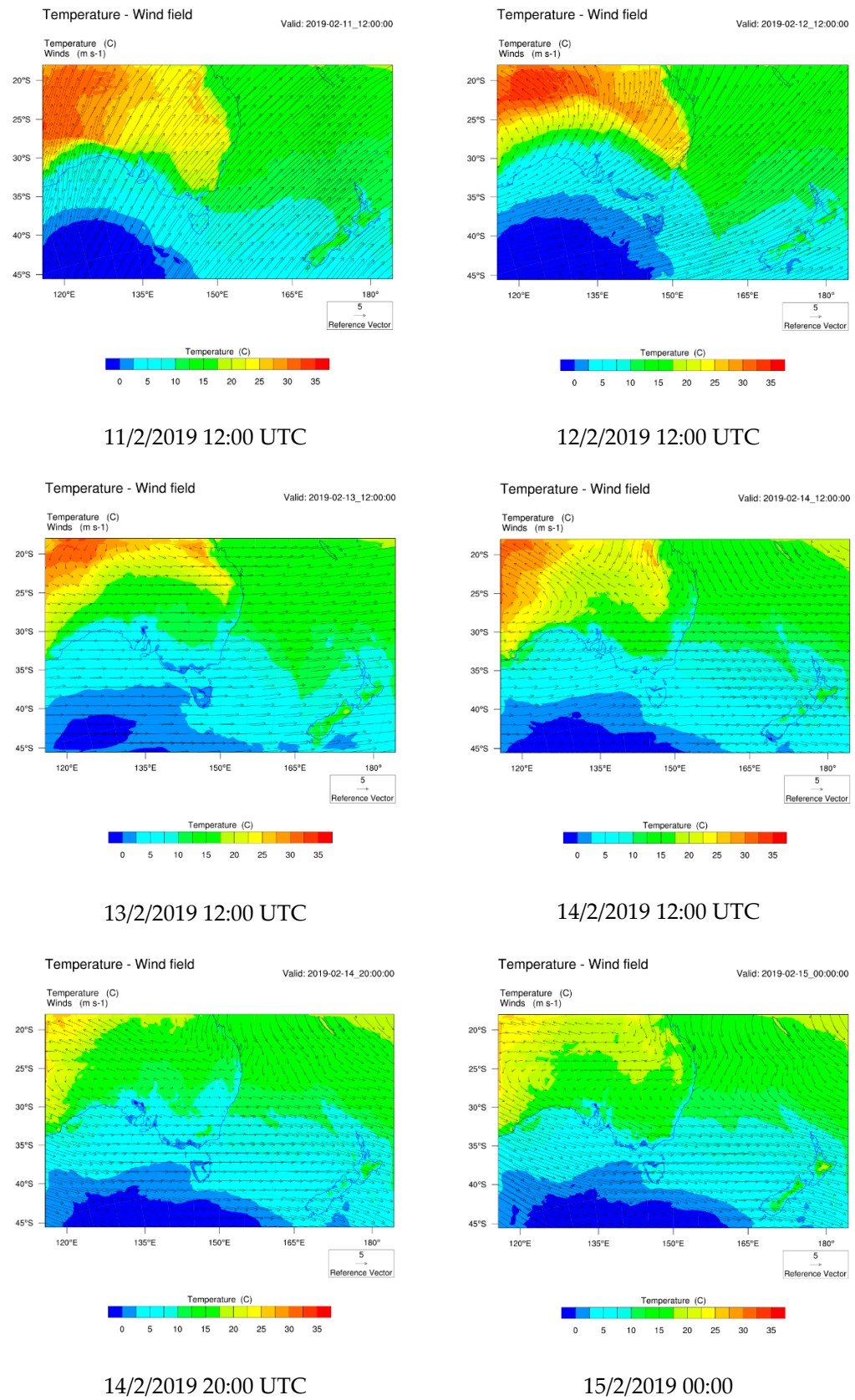

**Figure 5.** Surface temperature and wind field across the domain on 11 February 2019, 12 February 2019, 13 February 2019, and 14 February 2019 12:00 UTC. From 14 February 2019 12:00 UTC to 15 February 2019 00:00 UTC, the wind changed from easterly to north easterly at longitude 170 and beyond.

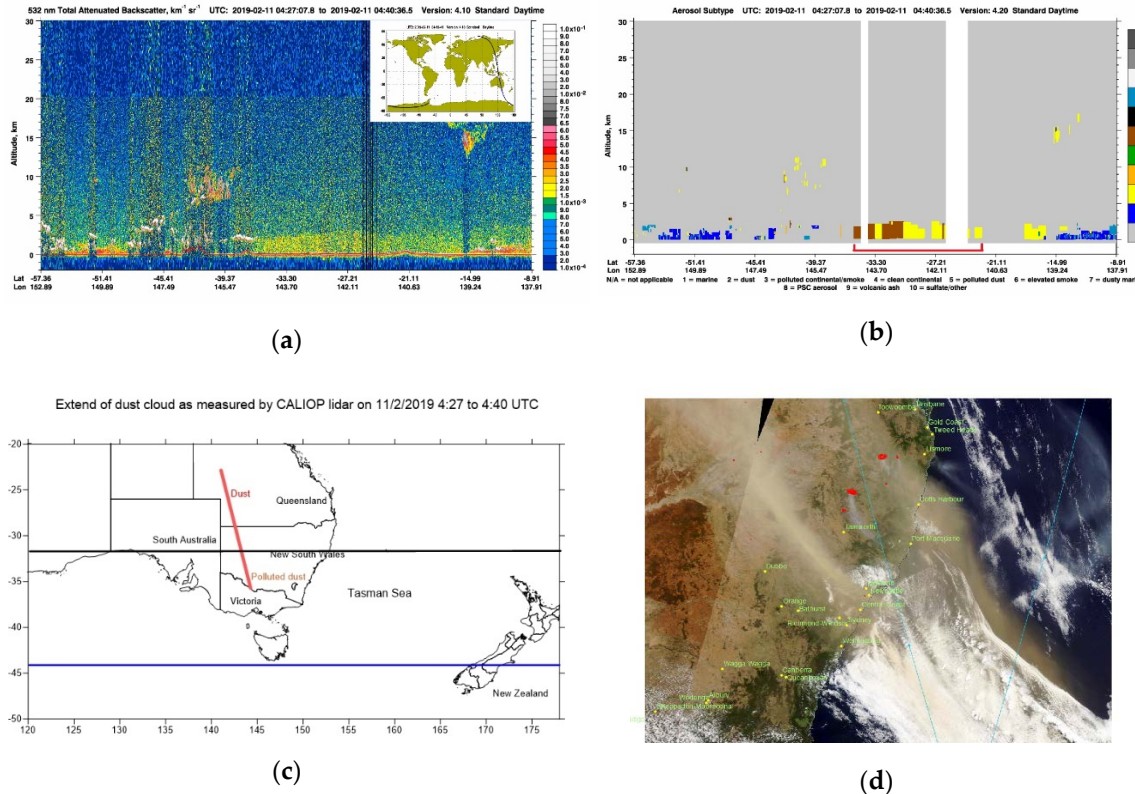

**Figure 6.** Extent of the dust cloud as detected by satellites: (**a**) LiDAR 532 nm total attenuated backscatter vertical structure on 11 February 2019 from 4:27 to 4:40 UTC with CALIPSO satellite path above central Australia; (**b**) aerosol subtype vertical profile as derived from CALIOP measurements. (**c**) Extent of the dust cloud (red line) as measured by CALIPSO satellite on 11 February 2019 4:27 to 4:40 UTC and cross sections of predicted $PM_{10}$ and wind profile (black and blue lines). (**d**) MODIS Terra satellite image of the dust event and wildfires in the New England area on 13 February 2019 and the locations of the main towns and cities in NSW.

The predicted vertical $PM_{10}$ and wind profiles allow us to understand the process of dispersion and evolution of the dust plumes from emission sources across the modelling domain. The $PM_{10}$ and wind profiles across the $-31.73°$ latitude horizontal line (black line) in Figure 6c at different times are shown in Figure 7. The south westerly wind carried the dust from the modelled sources in central Australia; the dust plume crossed the Great Dividing Range on north westerly winds (Figure 7a) over the coast to the Tasman Sea on south westerly winds (Figure 7b–d). The vertical wind (w-component) is small compared to the u- and v-components above this cross-section line.

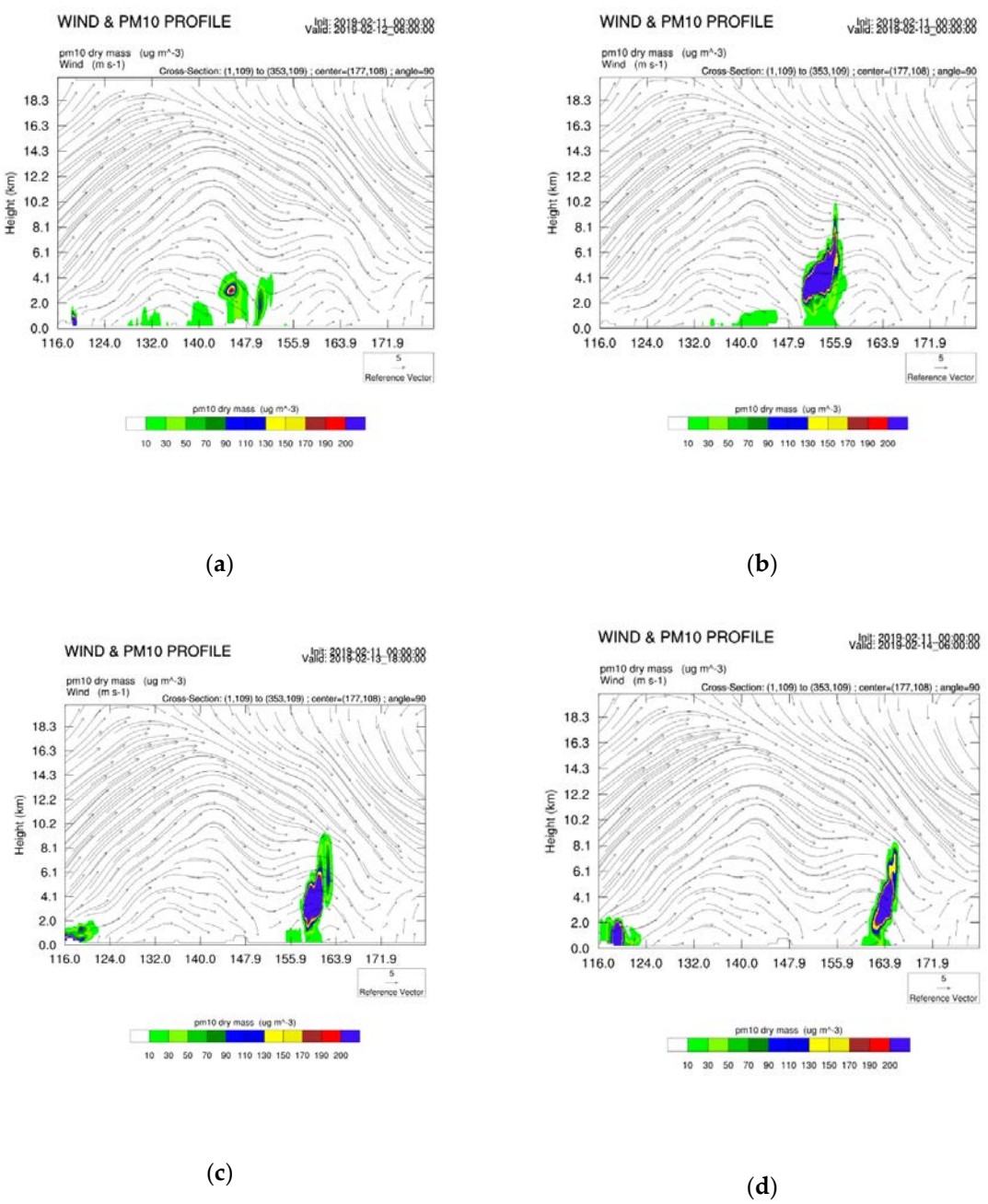

**Figure 7.** Predicted PM$_{10}$ and wind profile along the latitude −31.73° on (**a**) 12 February 2019 06:00, (**b**) 13 February 2018 00:00, (**c**) 13 February 2019 18:00, and (**d**) 14 February 2019 06:00 UTC. The vertical topographic outline at the bottom of the graphs shows the Great Dividing Range in the middle, with the western plain on the left and the Tasman Sea on the right.

*3.1. Dust Transport across the Tasman Sea to New Zealand*

Similar to the above vertical cross section of the dust source region in central Australia, shown as a horizontal black line in Figure 6c, the cross section below (blue line), in Figure 6c, at a latitude of −44.10°, over the town of Geraldine in New Zealand, shows that, on the 13 February 2019 at 00:00 UTC, the dust cloud first appeared along this cross section west of New Zealand's South Island. However, north and north easterly winds from the ground up to 10 km prevented the dust cloud from reaching the island until 16:00 UTC on the same day. The dust cloud stayed above the South Island from then

until 14 February 2019 from 04:00 to 06:00 UTC, when it intruded at ground level, causing elevated $PM_{10}$ concentration at Geraldine and other sites nearby (Figure 8).

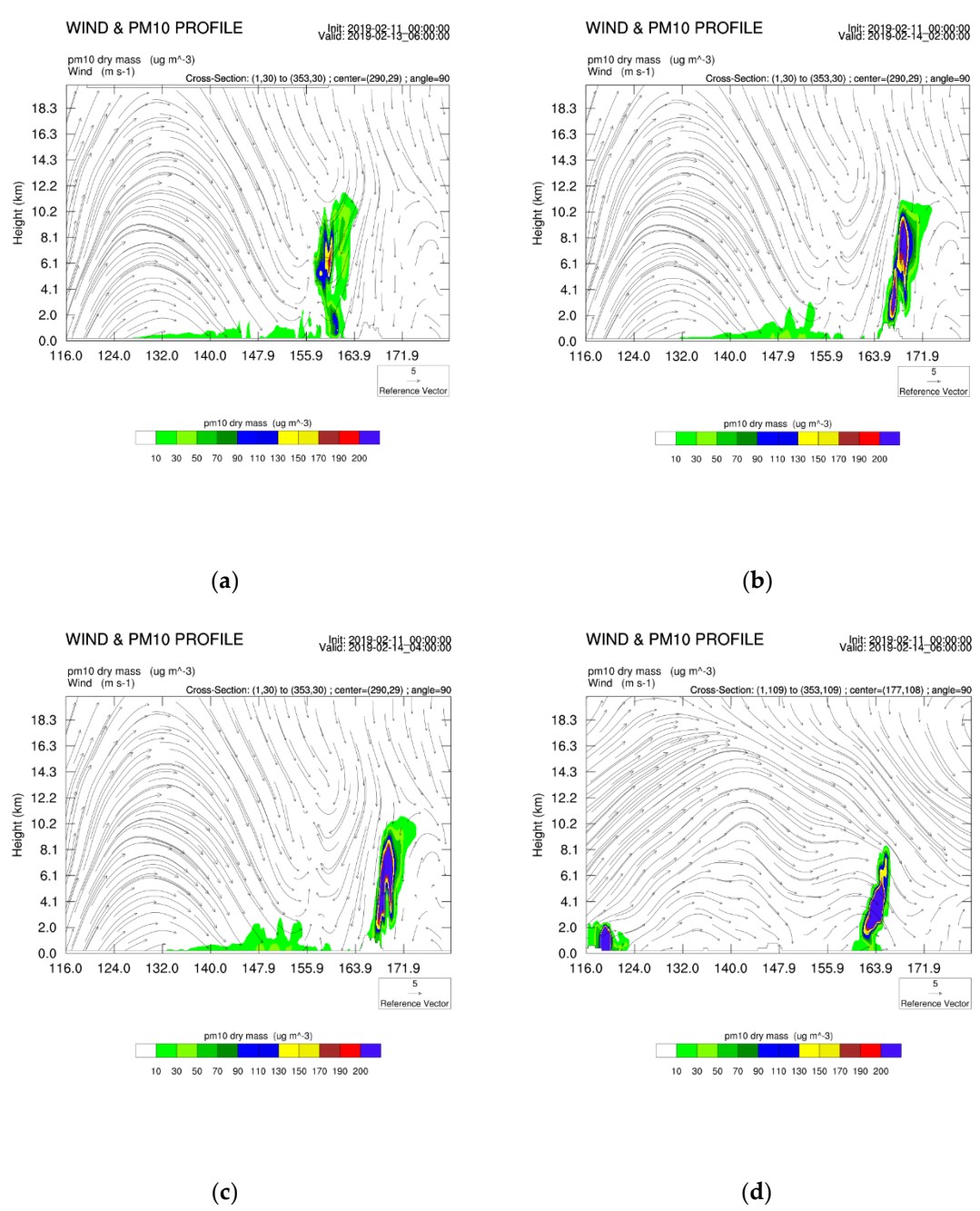

(**a**)  (**b**)

(**c**)  (**d**)

**Figure 8.** Predicted $PM_{10}$ and wind profile along the latitude −44.10° on (**a**) 13 February 2019 06:00, (**b**) 14 February 2018 02:00, (**c**) 14 February 2019 04:00, and (**d**) 14 February 2019 06:00 UTC. The vertical topographic outline at the bottom of the graphs shows the Tasman Sea and the South Island of New Zealand.

The progress of the dust and smoke plumes as viewed from above can also be seen in the AOD spatial pattern over time as revealed by the MODIS Terra/Aqua MO08-D3 and MYD08-E3 satellite products. The AOD from the Deep Blue algorithm for bright targets on land (such as deserts), and the Dark Target algorithm for dark targets (such as ocean) can be combined. These AODs are shown in Figure 9 for the 13 February 2019 from the Terra MOD08-D3 product and in Figure A3 of the

Appendix A for the period from 10 to 17 February 2019 from the Aqua MYD08-E3 product. The MYD08-E3 product provides the maximum and average of daily AOD for the 7-day period starting from 10 and ending on 17 February 2019, and shows the spatial pattern of AOD for this period to be a result of the dust transported in the February 2019 dust storm.

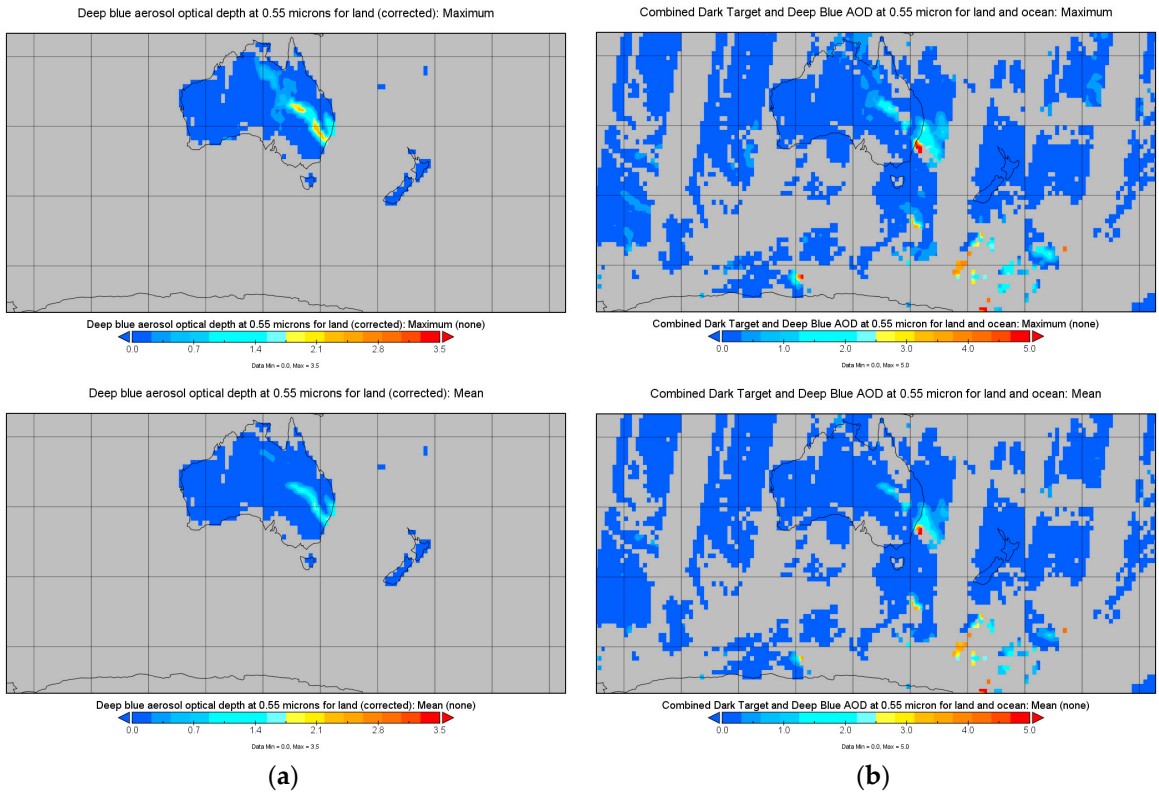

**Figure 9.** (**a**) Maximum and mean AOD Deep Blue Terra MOD08-D3 for 13 February 2019 and (**b**) combined maximum and mean AOD of Deep Blue (land) and Dark Target (ocean) from Terra MOD08-D3 for 13 February 2019.

At monitoring sites in the Canterbury region of the South Island of New Zealand, such as Geraldine, Timaru and Woolston (Christchurch), which are more than 100 km from each other, peak concentrations of $PM_{10}$ (and $PM_{2.5}$) were detected on 14 February 2019 05:00 UTC and at Rangiora, Kaiapoi about 2 h later. The locations of these air quality sites are shown in Figure 10a.

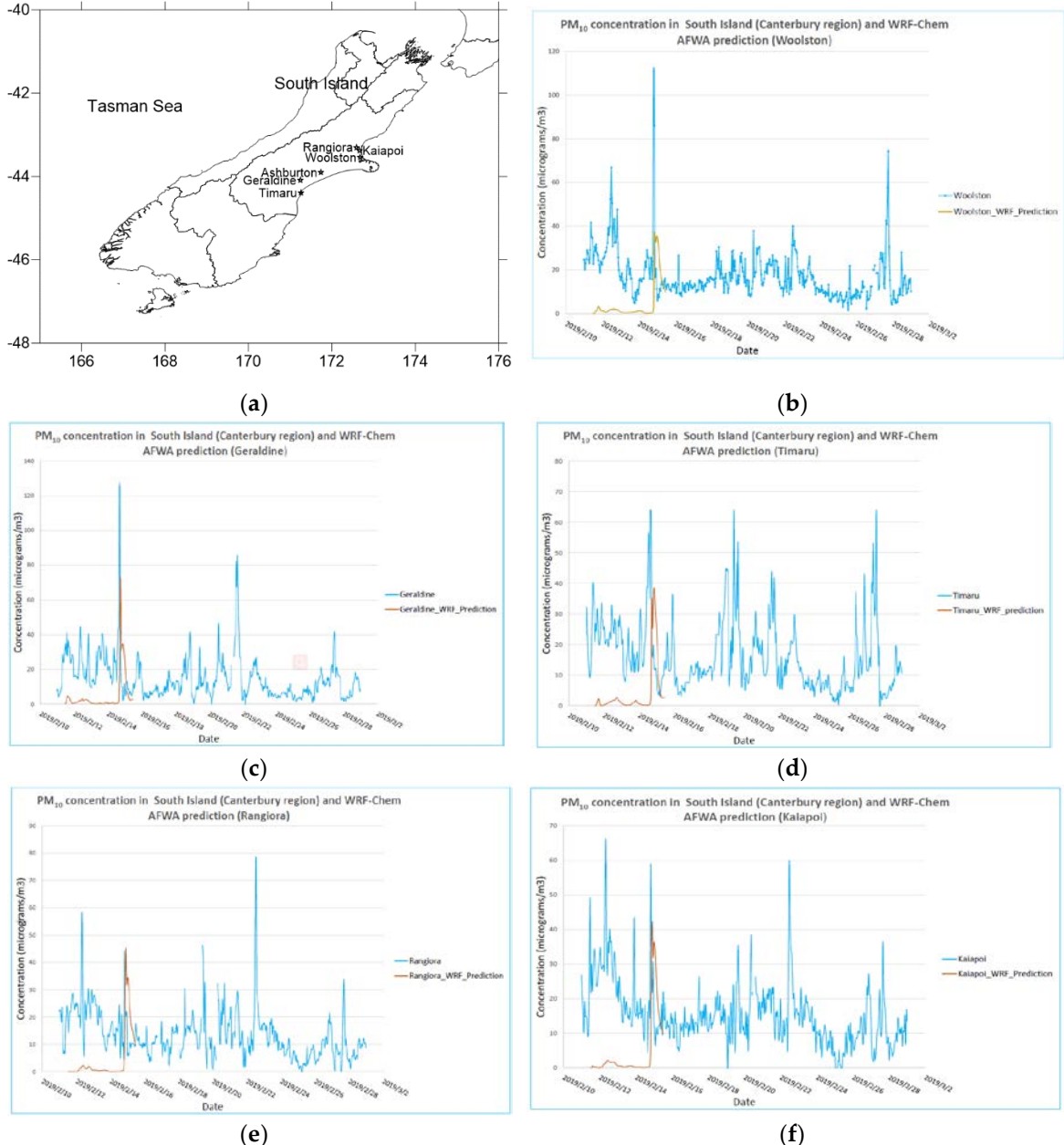

**Figure 10.** (**a**) Monitoring stations in the Canterbury area of the South Island, New Zealand, for 14 February 2019. (**b**–**f**) Predicted PM$_{10}$ from WRF-Chem with AFWA dust scheme (red line), FINN fire emission and MOZART and GOCART chemistry mechanism and observed PM$_{10}$ (blue line) at Woolston, Geraldine, Timaru, Rangiora and Kaiapoi.

The timing of predicted PM$_{10}$ peaks coincides with the time of the corresponding peaks at the monitoring sites, which are about 100 km apart. The maximum PM$_{10}$ concentrations of 127 μg/m$^3$ occurred at Woolston (14 February 2019 08:00 UTC), 110 μg/m$^3$ at Geraldine (14 February 2019 05:00 UTC), 64 μg/m$^3$ at Timaru (14 February 2019 05:00 UTC), 59 μg/m$^3$ at Kaiapoi (14 February 2019 08:00 UTC), and 44 μg/m$^3$ at Rangiora (14 February 2019 08:00 UTC). The average daily PM$_{10}$ concentration at these sites is about 20 μg/m$^3$. The maximum PM$_{10}$ concentrations, however, are underpredicted by WRF-Chem MOZART/GOCART chemistry. The GOCART AFWA dust model in WRF-Chem also varies greatly, depending on the value of the constants reflecting the local terrain conditions. Tuneable constants for dust emission in WRF-Chem with the dust option can be chosen so that the PM$_{10}$

prediction at the site can be compared with the observed data at the site. In Figure 10, the constant $C_{mb}$ is set at 7.0, rather than 2.61, as specified in the original Equation (8).

HYSPLIT forward trajectories of dust from the source area using the HYSPLIT matrix forward analysis, as shown in Figure 11 for trajectories starting from 11 February 2019 4:00 UTC, indicate that the dust was transported along two paths: the North East dust path and the Eastern dust path. The pattern of trajectories is similar when the starting time of release from sources is between 10 February 2019 06:00 and 11 February 2019 22:00 UTC.

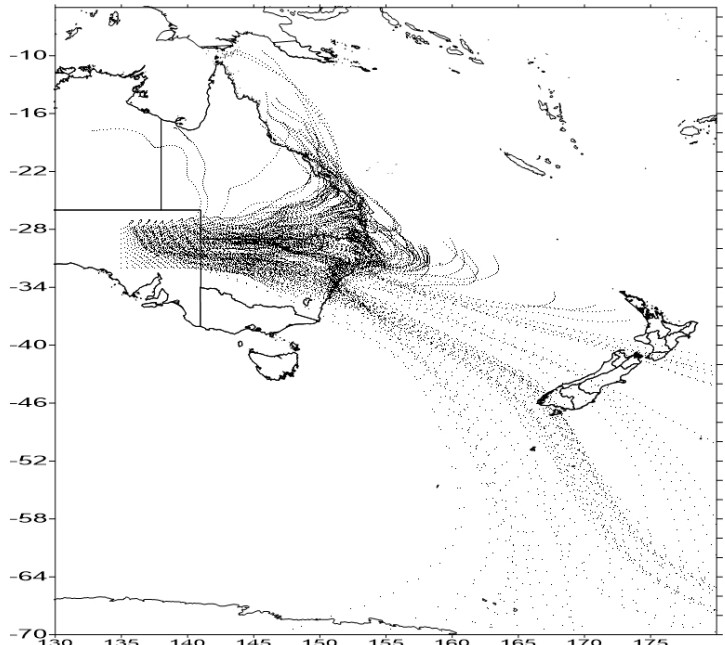

**Figure 11.** HYSPLIT forward trajectory ensemble analysis for 96 h at 10 m height from a source area in Central Australia on 11 February 2019 4 UTC.

The source area is bounded by the rectangular area, with a lower left coordinate of (135°, −32°) and an upper right coordinate of (145°, −28°), and each dust point source is at 50 m AGL (above ground level), separated by 0.5° (resolution) in latitude and longitude. This area encompasses known dust sources (Lake Eyre, South Simpson Desert, South Strzelecki Desert, Lakes Torrens and Gairdner region). The North East dust path carries dust from the northern part of Lake Eyre Basin (above latitude −30°) to Queensland and the Great Barrier Reef off the coast, while the Eastern dust path transports dust from southern part of Lake Eyre Basin and the Lakes Torrens and Gairdner region to the Tasman Sea, New Zealand and Antarctica.

CALIOP LiDAR onboard the CALIPSO satellite providing the vertical structure of backscatter and aerosol types on the 14 February 2019 near the west coast of the South Island of New Zealand shows dusty marine layers and layers of dust above ground. High above the fresh marine aerosol layers are dust, polluted dust and dusty marine layers. There are also thick smoke aerosol layers present in the LiDAR measurement (black in Figure 12b).

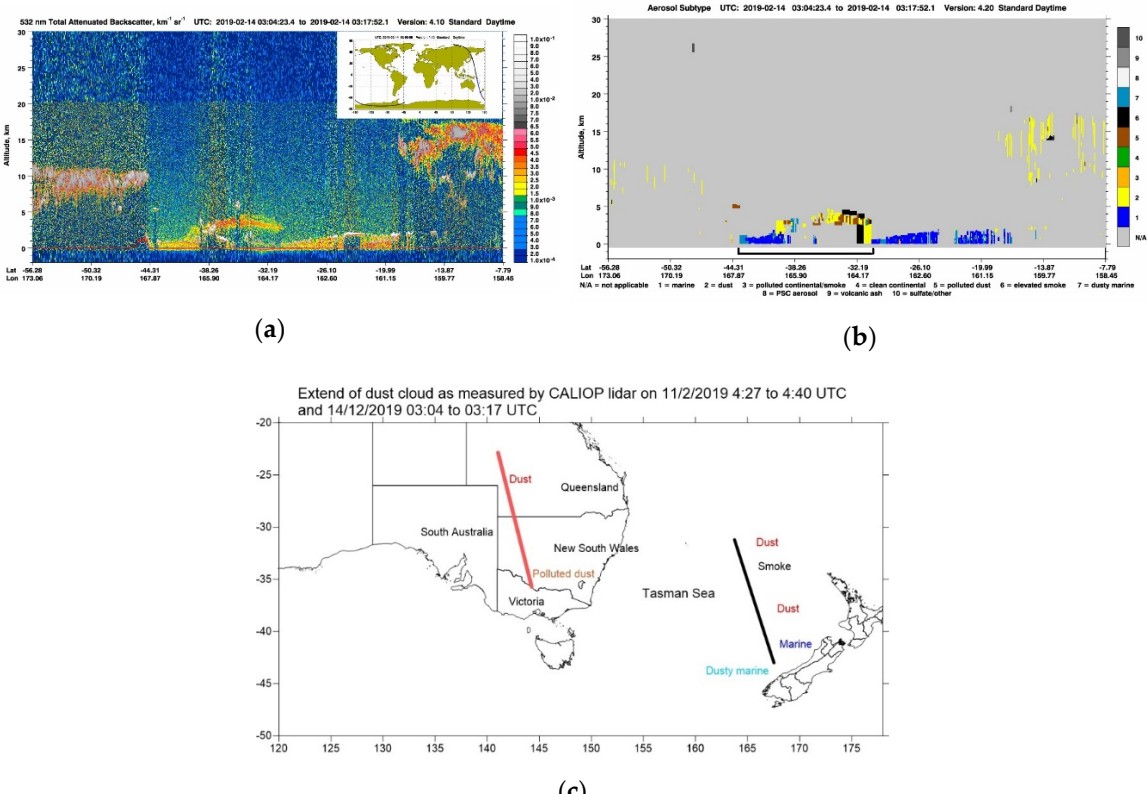

**Figure 12.** (**a**) LiDAR 532 nm total attenuated backscatter vertical structure on 14 February 2019 from 03:04 to 03:17 UTC with CALIPSO satellite path (inset) above the Tasman Sea near New Zealand. (**b**) Aerosol subtype vertical profile as derived from CALIOP measurements with dust as yellow and smoke as black. (**c**) Extent of aerosol layer clouds as measured by CALIPSO satellite on 11 February 2019 (red line) and on 14 February 2019 (black line). The labels correspond to aerosol types at the location indicated.

As shown in Figure 13a,b, the CALIOP LiDAR on the CALIPSO satellite, above the coast of eastern Australia on 13 February 2019, showed dust and smoke aerosol layers situated above the dusty marine and marine aerosol layers. The dust and smoke aerosol layers (from approximately 2 km to 5 km) above the dust-contaminated marine and fresh marine aerosol layers below (from just above sea level up to a height of 2 km) extend to about 1400 km long.

The pattern of dust and smoke from the CALIPSO vertical profile on the 13 February 2019 from 15:01 to 15:15 UTC can be compared with WRF-Chem vertical pattern of prediction of $PM_{10}$, showing dust of different sizes and total Black Carbon (both hydrophilic and hydrophobic Black Carbon) along the satellite path on 13 February 2019 15:00 UTC. Black Carbon (BC) is mostly indicative of combustible source origins, while $PM_{10}$ is a combination of both sources.

Figure 13d–f shows the transect of dust (bin size 4), total black carbon and $PM_{10}$ along the same CALIPSO satellite path of the coast of eastern Australia, shown as a blue line in Figure 13c. The WRF-Chem prediction of a concentrated layer of total black carbon was primarily caused by combustion sources such as the burning of biomass predicted at a height of approximately 4 km above the ground and centred around $-27.3°$ latitude and $155.3°$ longitude (Figure 13e), which corresponded to the observed smoke plume layer located above the same location at approximately the same height from 4 to 5 km (Figure 13b). Similarly, the prediction of the spatial vertical pattern of the $PM_{10}$ layer along the transect corresponds well with the CALIOP aerosol measurement pattern as shown in Figure 13b. The predicted pattern of dust for bin size 4 (3.0–6.0 μm) is largely similar to that of $PM_{10}$, which indicates that $PM_{10}$ consists mostly of dust, with only a minor contribution from fires.

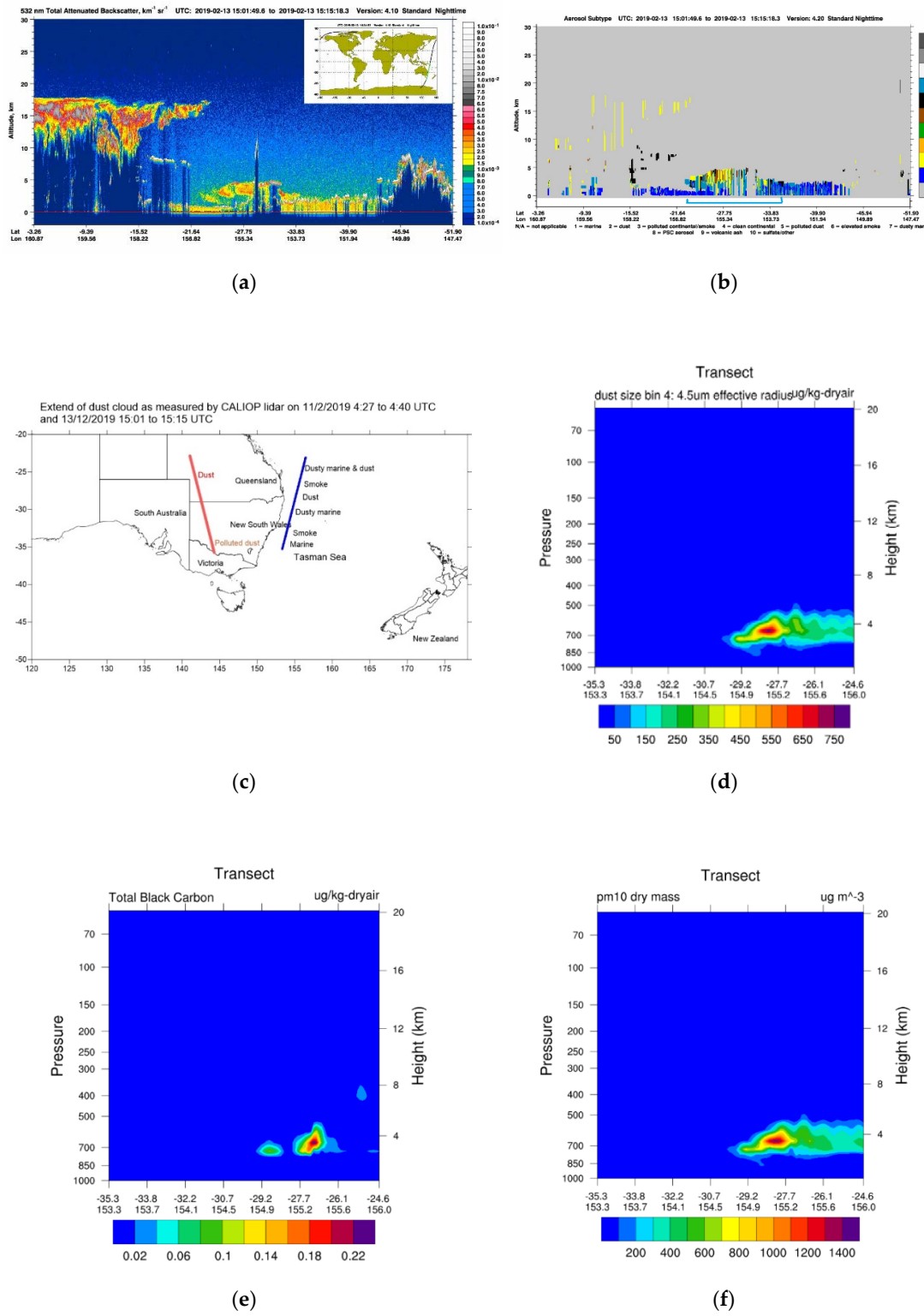

**Figure 13.** (**a**) LiDAR 532 nm total attenuated backscatter vertical structure on 13 February 2019 from 15:01 to 15:15 UTC with CALIPSO satellite path (inset) on 13 February 2019 above the coast of eastern Australia. (**b**) Aerosol subtype vertical profile as derived from CALIOP measurements. (**c**) Extent of dust clouds as measured by the CALIPSO satellite on 11 February 2019 (red line) and on 13 February 2019 (blue line). The labels correspond to aerosol types at the location indicated. (**d**–**f**) Transects of WRF-Chem predicted dust of size bin 4, total black carbon and $PM_{10}$ on 13 February 2019 15:00 UTC along the same CALIPSO satellite path.

### 3.2. Dust Transport to Antarctica

Using the HYSPLIT model, the forward trajectory analysis for 96 h from the source area bounded by the rectangular area with a lower left coordinate of (135°, −32°) and an upper right coordinate of (145°, −28°) at 50 m AGL shows that some of the dust trajectories not only reached New Zealand, but also passed over Antarctica and beyond to the south Pacific Ocean east of the South American coast (off Chile). Figure 14 shows the trajectories of transported dust from sources in Central Australia during the February 2019 dust storm.

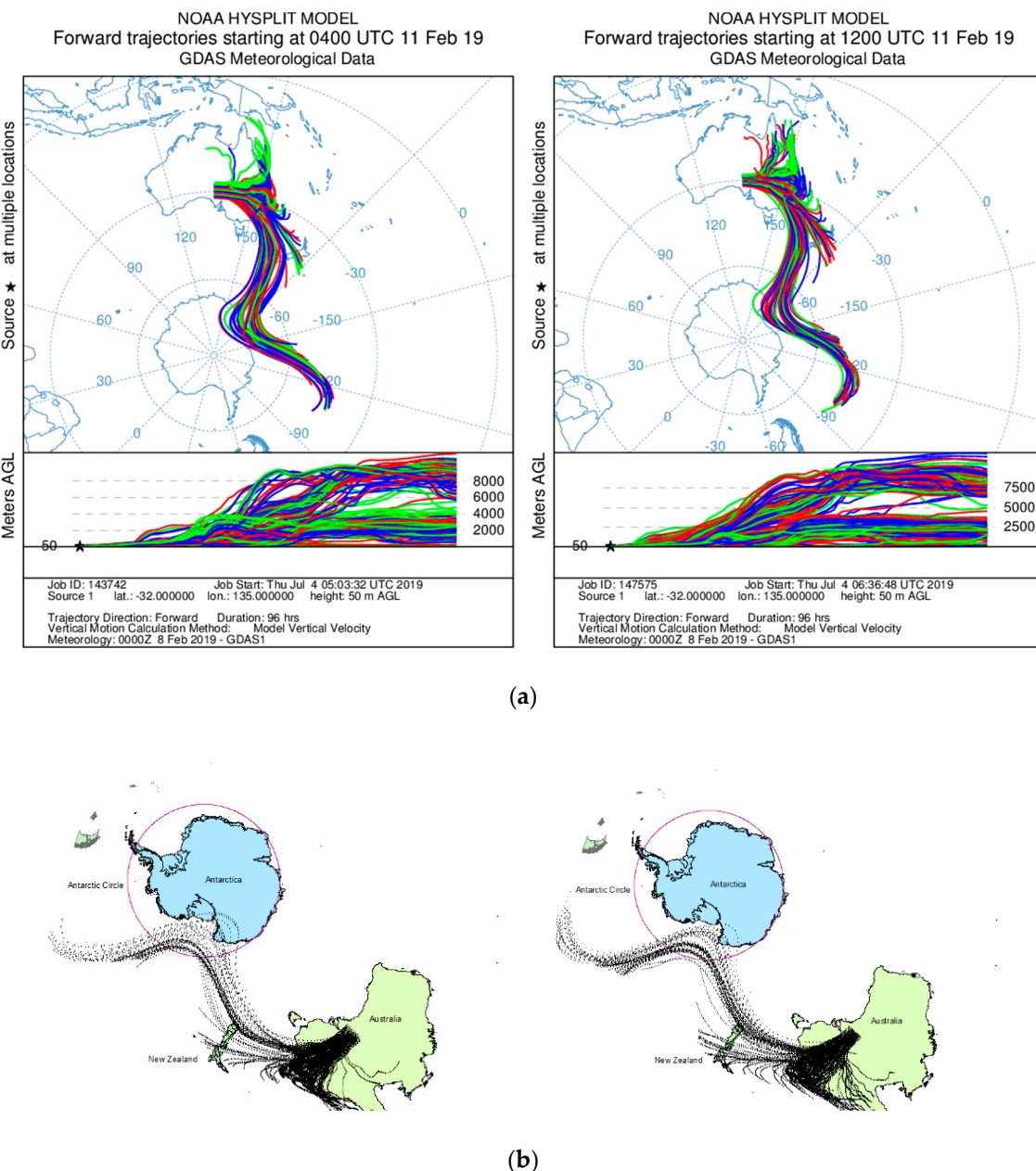

(**a**)

(**b**)

**Figure 14.** (**a**) NOAA HYSPLIT model matrix forward trajectory analysis from dust source areas in Central Australia with particle release times on 11 February 2019 4:00 UTC (left) and 12:00 UTC (right). (**b**) Same as (**a**), but using ArcGIS to display selected trajectories.

Satellite observation is used to correlate with the forward trajectory analysis. On the 15 February 2019, the dust from the event that started on 13 February 2019 passed through New Zealand and reached the edge of Antarctica, as detected by the MODIS Terra/Aqua sensors. The CO emission from

bushfires was still detected as a CO column concentration of about 90 ppb in the New England area of northern NSW (green colour in Figures 15 and 16).

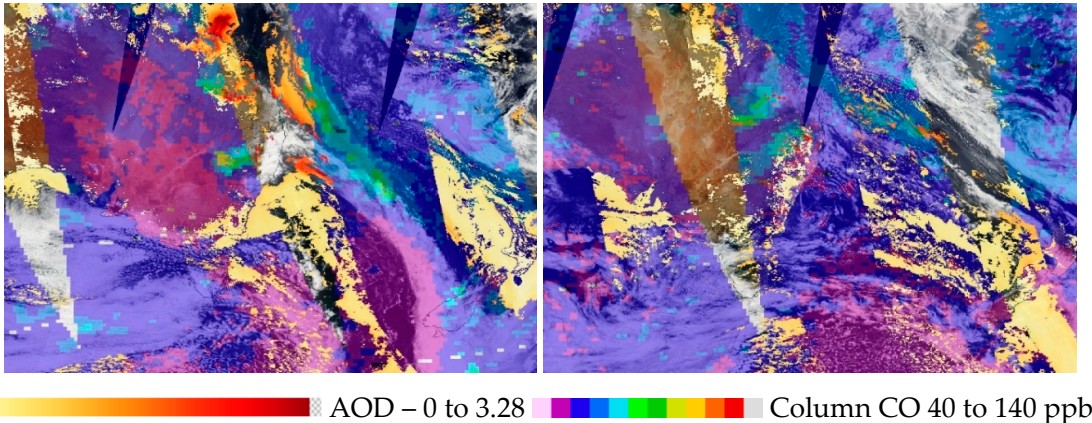

**Figure 15.** AOD (**yellow and red**) and CO column concentration Aqua/Airs (**pink and green**) over the Tasman sea, as revealed by MODIS on 14 February 2019 (**left**) and 15 February 2019 (**right**).

Figure 16 shows the AOD from Aqua/Modis 3 km land and ocean for dust and smoke particles, and carbon monoxide (CO) from Aqua/Airs due to biomass fires from 13 February 2019 to 14 February 2019. They show the progress of the dust plumes, and that New Zealand (both the North and South Islands) is slightly affected by dust (yellow and red). The dust column plume then veers towards Antarctica.

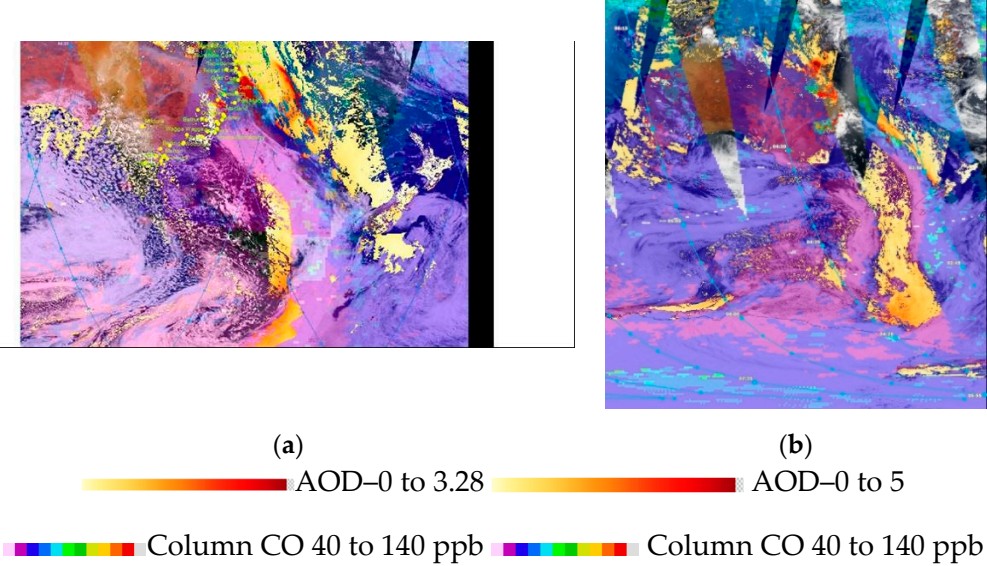

(**a**)                                      (**b**)

**Figure 16.** The AOD from Aqua/Modis and the CO column Aqua/Airs daytime concentration on 13 February 2019, as well as the location of towns and cities on the eastern seaboard of Australia (**a**); and the AOD from Terra/Modis and the CO column Aqua/Airs daytime concentration on 14 February 2019 (**b**). The pinkish and greenish colours are CO (carbon monoxide) column concentration.

The overall extent of AOD and the daily AOD average over the regions of Australia, New Zealand and Antarctica on the 13 and 14 February 2019 are shown in Figure A4 of the Appendix A. The AOD over Antarctica is a result of several different aerosol types: marine aerosols, and dust and fire emission aerosols. CALIOP LiDAR data is used in order to resolve the origin of AOD above Antarctica. Figure 17 shows the CALIOP LiDAR vertical structure of backscatter and aerosol type as measured by the CALIPSO satellite on 15 December 2019.

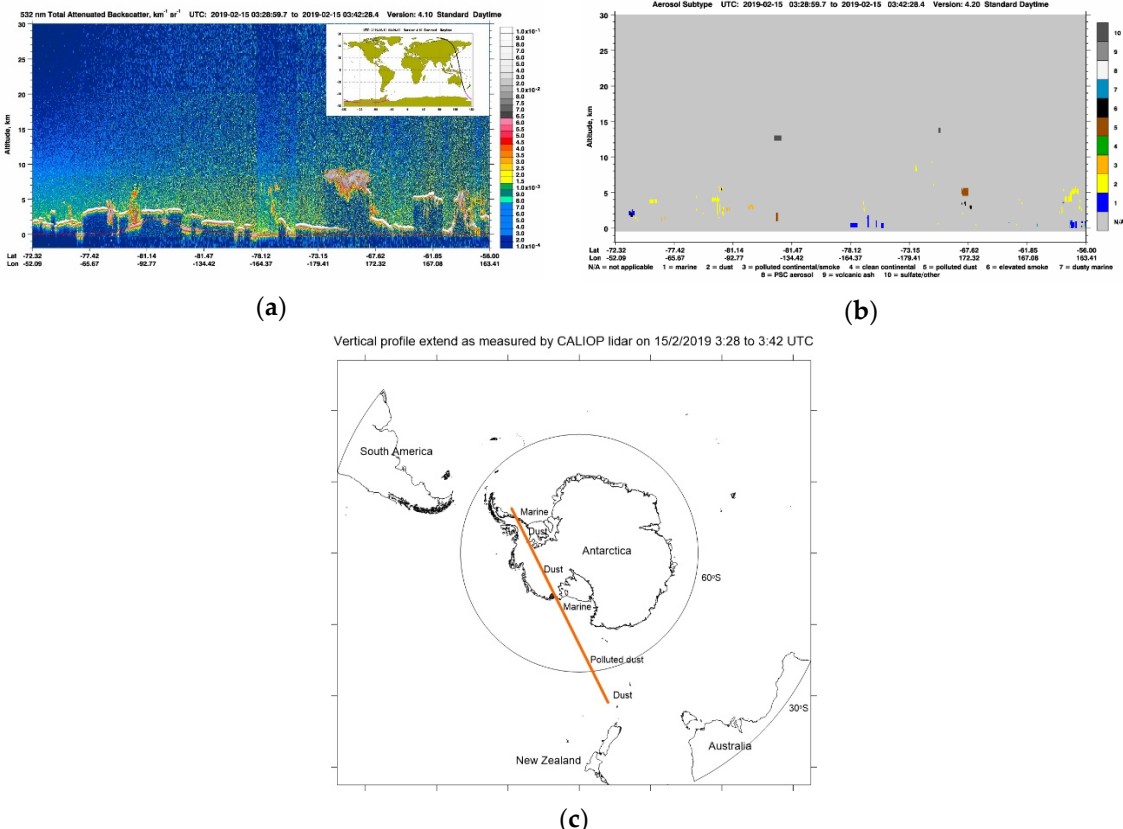

**Figure 17.** (**a**) LiDAR 532 nm total attenuated backscatter vertical structure on 15 February 2019 from 03:42 to 03:42 UTC with CALIPSO satellite path (inset) above Antarctica. (**b**) Aerosol subtype vertical profile as derived from CALIOP measurements (**c**) Extent of vertical profile as measured by CALIPSO satellite on 14 February 2019 (red line). The labels correspond to aerosol types at the location indicated.

It is clear from CALIOP LiDAR measurements that the aerosols above Antarctica are of dust origin (dust and polluted dust types) at heights from 2 km to 6 km, with marine aerosols confined to the layer 0 to 2 km above ground level. These high dust layers above Antarctica indicate long-range transport of dust from Australia. As for smoke particles due to wildfires in northern NSW, Australia, they were transported to the Tasman Sea (at high latitude and altitude) and did not affect the ground concentration in the South Island of New Zealand. However, scattered smoke layers at very high altitude (from above 10 km up to 15 km) were detected, as shown in Figure 17, above Antarctica, and they could have had their origins in large fires and biomass burnings in Australasian regions of Indonesia and northern Australia.

## 4. Discussion

Australia is a dry continent, and wildfires occur in many parts of Australia, mainly in northern Australia, and especially during the dry and hot period of summer. In addition, frequent dust storms from deserts in central Australia and agricultural land where annual rainfall is below 500 mm can cause dust to be transported under favourable meteorological conditions all the way to the east coast, and beyond to the Tasman sea, New Zealand, and as far as Antarctica. This transport of dust along the south east dust pathway has been occurring for 350,000 years [14]. In glacial periods, dust transport increases. In three out of four cores analysed by Hesse 1994 [4], sediment accumulation rates had fallen since peaking 130,000 to180,000 years ago. De Deckker et al. 2010 [19] reported that while Australia is not currently in a glacial period, its environment has been becoming increasingly dry over the last 6000 years.

Marx et al. 2018 [11] reported that the global conversion of land to agriculture over the last 250 years has resulted in increased dust sources. Ginoux et al. 2012 [40] estimated that 25% of global dust emission was from areas associated with anthropogenic activities, such as agriculture. In the case of eastern Australia, dust sources are a mix of natural and anthropogenic sources. Natural sources in central Australia like dry lakes and floodplains are driven by the El Niño–Southern Oscillation (ENSO) [51]. During wet La Niña years, sediment is transported to the topographically low areas of Central Australia. During the El Niño dry years, dust is emitted and transported. Marx et al. 2018 [11] also reported that dust concentration at Hokitika, South Island, New Zealand increased between 1991 and 2001, and with the current Southern Oscillation Index (SOI) having been mostly negative since 2014 (their Figure 9), this was also contributing to the current dust transport. Prior to 1991, they reported dust deposition in the Australian Snowy Mountains being about three times higher than post 1990 (their Figure 23).

Brahney et al. 2019 [52], in their study of dust deposition at two alpine lakes in Kahurangi National Park, South Island of New Zealand, found by using geochemical proxies that dust deposition began to increase around 1900, with the greatest deposition rates occurring from ~1920 to ~1990, and appeared to have declined in subsequent decades. This change in dust deposition in New Zealand lakes is consistent with dust records from the Antarctic Ice Sheet and Eastern Australia. They found that this temporal change in dust deposition rate followed the temporal pattern of land use in south and central Australia during the twentieth century. In addition, they suggested that land clearing and agricultural practices causing a significant increase in soil erosion, especially in the early 20th century, was the cause of increased dust deposition in New Zealand lakes.

This study shows that dust is also transported to Antarctica, as was also shown by De Deckker et al. 2010 [19], who reported an event starting on 1 April 2008 in central and south eastern Australia that travelled to Antarctica over the next 11 days. This process has been going on for thousands of years. Dust in ice cores dating back 170,000 years shows evidence of dust being transported from the south east of the Australian continent [19].

O'Loingsigh et al. 2017 [17] showed that the Eastern dust pathway transport of dust to the coast from sources in central Australia, as described in their study, was associated with dust export to New Zealand and dust deposition over the Tasman Sea. In our study, the dust event of February 2019 followed the eastern pathway to the coast, and our results confirm that dust from this event did indeed cross the Tasman Sea and reach as far as New Zealand and beyond, including Antarctica. The long-range transport of dust from the deserts and dry plains in central Australia and western New South Wales to the Tasman Sea and beyond is also made possible by the lifting up of the dust cloud after crossing the elevated Great Dividing Range, which is about 3700 km long, roughly paralleling the east coast of Australia and extending from Cape York peninsular in Queensland down through New South Wales and Victoria. In New South Wales, the average height of the highland is about 1000 m, rising to a peak of 2228 m at Mount Kosciuszko in the Australian Alps near the New South Wales–Victoria border. Xu et al. 2018 [53], in their study of global dust transport from the Tibetan Plateau (TP), found that dust load and downstream dust flux were significantly higher in the upper troposphere (above ~8 km) with respect to those over other major dust sources in the Northern Hemisphere. This enables the long-range transport of dust around the Northern Hemisphere, as the TP acts as a channel for transporting dust from the lower atmosphere to the upper troposphere. The Great Dividing Range is not as high as the TP, but the lifting role it plays is similar.

Ginoux et al. 2004 [6], on the basis of their modelling study on the global distribution of aeolian dust from 1981 to 1996 with the Global Ozone Chemistry Aerosol Radiation and Transport (GOCART) model, estimated that the North African deserts (Sahara and Sahel) contribute about 65% of the global emissions of dust. Compared to other regions, Australia's annual average budget of dust emission (from 1981 to 1996) is 61 Tg year$^{-1}$, which is less than that of North Africa (1430) and Asia (496), but higher than that of South America (49), South Africa (22), and North America (9). North African dust sources also contribute to about half of the dust aerosols to the oceans, globally [23]—mainly to the

Atlantic Ocean. Australian dust storms from Central Australian deserts are mostly deposited in the Southern Ocean and Tasman Sea. In this study, dust from the event that took place in the period 11–15 February 2019 was shown to have been transported by predominantly south westerly from the emissive dust sources in the centre of Australia to the east coast, and then north easterly to the Tasman Sea. After encountering easterly winds, the dust was then transported southward, reaching New Zealand and Antarctica.

Compared to other studies of dust transport in the northern hemisphere, such as in Asia, where the long-range dust transport from the Taklimakan and Gobi deserts usually follows the eastward paths by means of prevailing midlatitude westerlies across China, reaching the Korean Peninsula and Japan, and occasionally being transported across the Pacific to North America [54]. The dust transport from the Central Australian desert and arid regions also follows eastward patterns by means of prevailing south westerlies or westerlies toward the Tasman Sea, New Zealand and beyond, occasionally even to Antarctica. Based on MISR (Multiangle Imaging Spectroradiometer) and HYSPLIT forward trajectory analysis, Yu et al. 2019 [54] studied the trans-Pacific dust transport from the Asian deserts to North America and found that Taklimakan dust predominantly affected areas south of 50° N in North America, while Gobi dust primarily affected areas north of 50° N. Zhang, Tong et al. 2016 [55], in their study of dust source identification in East Asia between 2010 and 2015 using MODIS data and HYSPLIT trajectory analysis, showed that dust from central Asian deserts such as the Taklimakan Desert and the Gobi Desert can be transported northwards by the Mongolian Cyclone to the Far East region, and even as far as the Arctic Circle.

Li et al. 2008 [56] studied the distribution, transport and deposition of dust in the Southern Ocean and Antarctica from dust sources in South America, Australia, and the Northern Hemisphere using GFDL Atmospheric General Circulation Model AM2 for a 20-year period (1999–1998). They found that the majority of dust transported to the Southern Ocean came from Australia (120 Tg/year), followed by Patagonia in South America (38 Tg/year) and inter-hemispheric transport from the Northern Hemisphere (31 Tg/year). A small fraction of this dust is deposited in the Southern Ocean and Antarctica (7 Tg/year). Our study of the February 2019 dust storm in eastern Australia shows that dust from this event was transported to Antarctica.

As Yu et al. 2019 [54] noted, when using the tracer mode in trajectory Lagrangian modelling analysis such as HYSPLIT, gravitational settling and wet deposition processes are ignored; therefore, the long-range transport of dust is usually overestimated compared to the actual range. In this study, we primarily used the Eulerian dispersion WRF-Chem model, which takes into account wet and dry deposition, but Lagrangian HYSPLIT was also used to illustrate the ensemble pathways of dust trajectories as likely dust transport routes to complement and support our understanding of the dispersion process predicted by WRF-Chem.

As for the dust model in the WRF-Chem model, there are limitations to the current WRF-Chem dust schemes based on topographic soil erodibility, which is confined mostly to the central Australian arid region. Rather than static erodibility, defined as soil erosion efficiency under a given meteorological forcing (i.e., wind speed), soil erodibility changes dynamically depending on the local environmental conditions, including land use, land cover type, and soil moisture [57]. In our study, soil moisture was provided by WRF-Chem meteorological component. Observation and satellite data such as MODIS have been used to calibrate this dynamic erodibility in the dust source region. However, land use and land cover (or green fraction) are based on static maps. Agricultural areas, such as the western Riverina and Mallee regions in the border area between New South Wales and Victoria, are not properly accounted for in the topographic soil erodibility map. The Riverina and Mallee regions have also been identified as dust sources [17,58,59] when using the rainfall data from the Australian Bureau of Meteorology (BOM) and land cover data derived from MODIS data to account for seasonal changes in soil erodibility and dust emission in order to study the dust climatology over a ten-year period in western New South Wales (NSW), Australia.

Similarly, Ginoux et al. 2012 [40] used $0.1° \times 0.1°$ gridded satellite data from MODIS Deep Blue (MODIS DB) Level 2 from 2003 to 2009 to detect and classify natural and anthropogenic dust sources for different seasons. The inclusion of anthropogenic activities such as land clearing, agricultural activities and grazing and ephemeral stream, lakes and rivers in arid region (such as Lake Eyre Basin in Australia) allowed them to assess anthropogenic and hydrological impacts on dust emission at a global scale. The largest source with the highest annual frequency over the Australian continent was in the channel country, specifically at the mouth of the Warburton River feeding North Lake Eyre. Some of the lakes in Lake Eyre Basin are active dust sources all year long, while others are active for one or two seasons, or not active for all seasons. The inclusion of MODIS data to improve the dust source function (Equation (2)) by detecting more hot spots or sources may have resulted in better prediction of dust concentration and deposition in our study. However, when applying the new Ginoux source function in West Asia, Nabavi et al. 2017 [43] found that it did not provide a significant improvement in accuracy for WRF-Chem predictions.

The use of satellite data can result in false identification of dust sources where persistent or seasonal long-range transport and deposition of dust or biomass burning occurs. To eliminate this false identification, Parajuli et al. 2014 [57] used the correlation between MODIS Deep Blue AOD and wind speed as a proxy for erodibility over an area. Erodibility as the maximum correlation of AOD and wind speed during the driest season is used and any correlation below this maximum reflects variation in environment condition such as soil moisture, vegetation cover. The use of such dynamic erodibility proxy can improve the prediction of dust concentration and AOD in the current dust scheme in the WRF-Chem model.

In our current study on the dust event of February 2019, the dust emission was mostly confined to the Lake Eyre Basin, and dust was transported across New South Wales to the east coast of Australia. Other Australian studies [51,60–62] have highlighted the importance of Central Australia as a dust source area. Zhang et al. 2016 [55], in their study of the identification of dust sources and hotspots in East Asia during the 2000–2015 period, analysed more than 2000 MODIS images of dust storm events and HYSPLIT back trajectory analyses, finding that the dominant dust sources are sandy lands and lake beds, rather than sandy and stone deserts, with increased frequencies in summer and autumn. The Lake Eyre Basin, whose ephemeral lakes and streams are fed with flood water from Queensland, was identified by Ginoux et al. 2012 as the source of most dust storms in east Australia during the dry and drought period.

Zhang et al. 2016 [55] also showed that changes in land use associated with anthropogenic activities such as mining and excessive exploitation of water resources are one of the major factors leading to the expansion of dust source regions, especially for the northeastern part of the Taklimakan desert.

Our future study of dust transports of dust storm events from Central and Eastern Australia will make use of seasonal soil erodibility maps derived from satellite data in WRF-Chem to improve the prediction of dust transport and concentration over the modelling domain. This erodibility map will include the Riverina–Mallee region as a significant dust source. The results of this study could then be compared with the current Ginoux dust source function to determine whether the new Ginoux source function is better for dust prediction.

Anisimov et al. 2018 [63], in their study of haboob (desert dust storm) over the Arabian Peninsula using WRF-Chem 3.7.1 with the GOCART dust emission scheme and MOSAIC aerosol chemistry, found that WRF-Chem underestimated the $PM_{10}$ mass concentration by a factor of nearly 2. They suggested that the current dust GOCART parameterisation for the size distribution of emitted dust underestimates the number of large particles under strong wind conditions. Does et al. 2018 [24] found that even giant particles (>75 μm) were transported long-range from the Sahara Desert to the Atlantic Ocean.

In our study, using the WRF-Chem GOCART-AFWA emission dust scheme, we also found that the predicted $PM_{10}$ using the original constant $C_{mb} = 2.61$ in the horizontal saltation flux equation instead of 1 from Marticorena and Bergametti 1995 [46], as implemented in WRF-Chem, still underpredicted the $PM_{10}$. When this tuneable constant was increased to 7, the result matched the observation better. Eltahan et al. 2018 [31] found that using a tuneable constant $C_{mb} = 6$ resulted in AOD prediction matching better with the AOD as measured by AERONET network over Cairo during the 31 March 2013 dust storm in Egypt.

## 5. Conclusions

In this study, we conducted simulations of dust emission and wildfire smoke transport from Australia to the Tasman Sea, New Zealand, and beyond to Antarctica from 11 to 15 February 2019. Such long-range transport and deposition of dust has been occurring for hundreds of thousands of years, as evidenced by the detection of dust in deep sea sediment cores in the Tasman Sea and ice cores in New Zealand and Antarctica. The 11 to 15 February 2019 event sits within a context of dust emissions being lower than in previous glacial periods, and lower than the 1900 to 1990 period, but increasing since 1991 [11]. The increase observed over the last 30 years has been a function of a series of dry periods.

The simulation using WRF-Chem and observed data on the ground and from satellites showed that long-range transport of dust has affected air quality, not only on the eastern seaboard of Australia, but also across the Tasman Sea on the South Island of New Zealand, as detected by monitoring stations in the Canterbury region. This is due to the intrusion to ground level from above of long-range transported dust layers.

In northern NSW, there were wildfires during this period. Simulations of both dust and wildfires emission in WRF-Chem and observation data from CALIOP LiDAR showed that aerosols of both sources were carried to the Tasman Sea, but the aerosol clouds contained more dust aerosols than aerosols from the burning of biomass.

There is evidence of dust transport to Antarctica at high altitude, as measured by MODIS satellite AOD and CALIPSO aerosol vertical structure from CALIOP LiDAR above Antarctica from the storm in Australia during the 11–15 February period. The transport of dust from Australia to New Zealand and beyond is also due to the current drought in eastern Australia. This continues a pattern that has occurred for thousands of years, as reported in many previous studies, including studies of lake sediment and ice core data from New Zealand and Antarctica.

This study of the dust storm in February 2019 contributes to the understanding of dispersion and long-range transport of dust from this event and confirms that dust following the eastern pathway from central Australia to the coast is able to reach New Zealand and Antarctica.

**Author Contributions:** Conceptualization, H.D.N., M.R., J.L.; methodology, H.D.N., J.L.; data procurement: H.D.N, D.S., J.L.; formal analysis, H.D.N.; investigation, H.D.N., J.L.; writing—original draft preparation, H.D.N., J.L.; visualization, H.D.N.; supervision, H.D.N., M.R.; project administration, M.R.

**Funding:** This research received no external funding.

**Acknowledgments:** Analyses and visualizations, where they are indicated in the paper, were produced with the Giovanni online data system, developed and maintained by the NASA GES DISC, and CALIPSO satellite products from NASA Langley Research Centre. We also acknowledge the NOAA Air Resources Laboratory (ARL) for the provision of the HYSPLIT transport and dispersion model and/or READY website (http://www.ready.noaa.gov) and Environment Canterbury Regional Council for the air quality data used in this publication.

**Conflicts of Interest:** The authors declare no conflict of interest.

## Appendix A

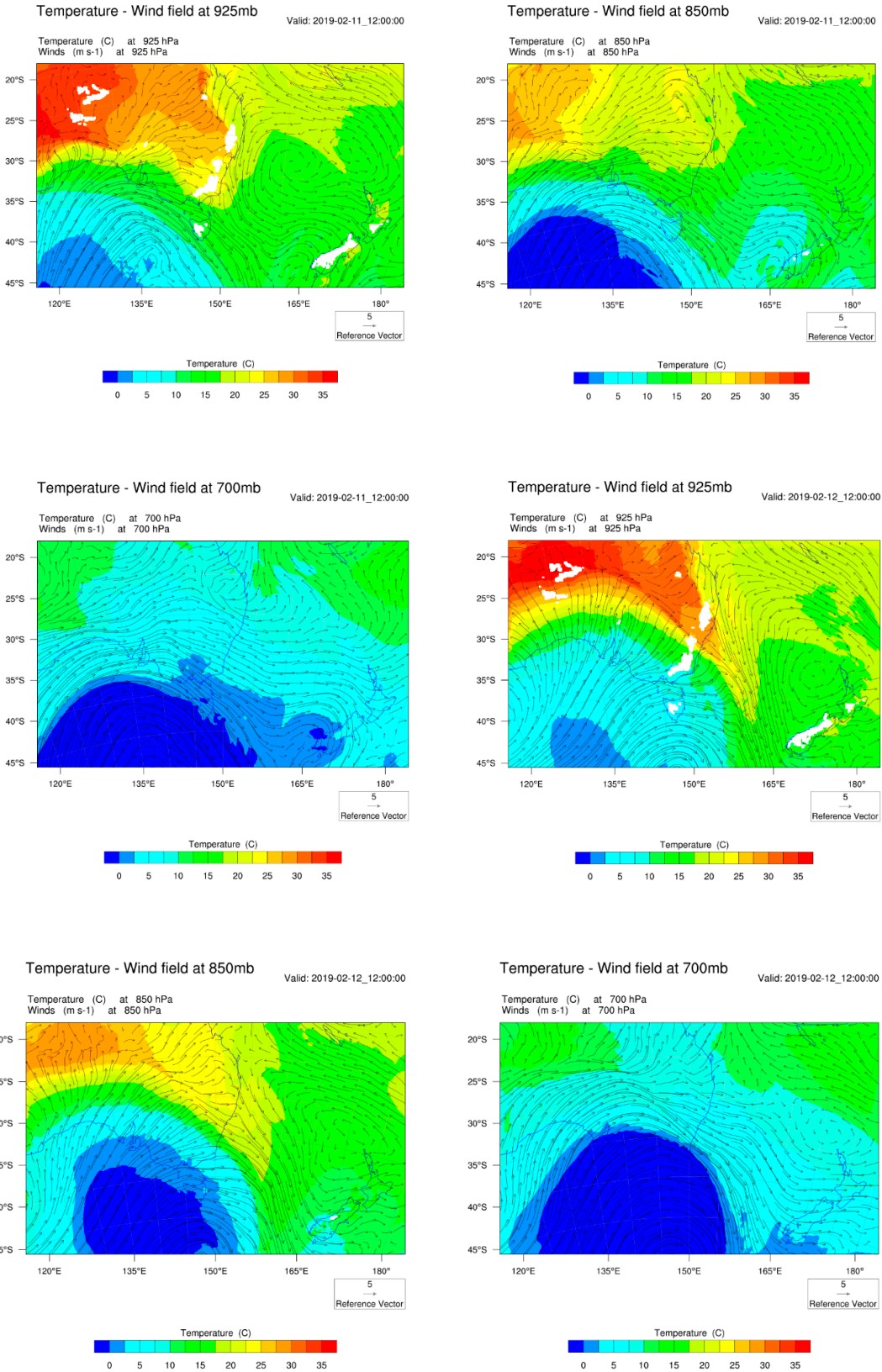

**Figure A1.** Temperature and wind field at 950 mb, 850 mb and 700 mb on 11 February 2019 12:00 UTC and 12 February 2019 12:00 UTC.

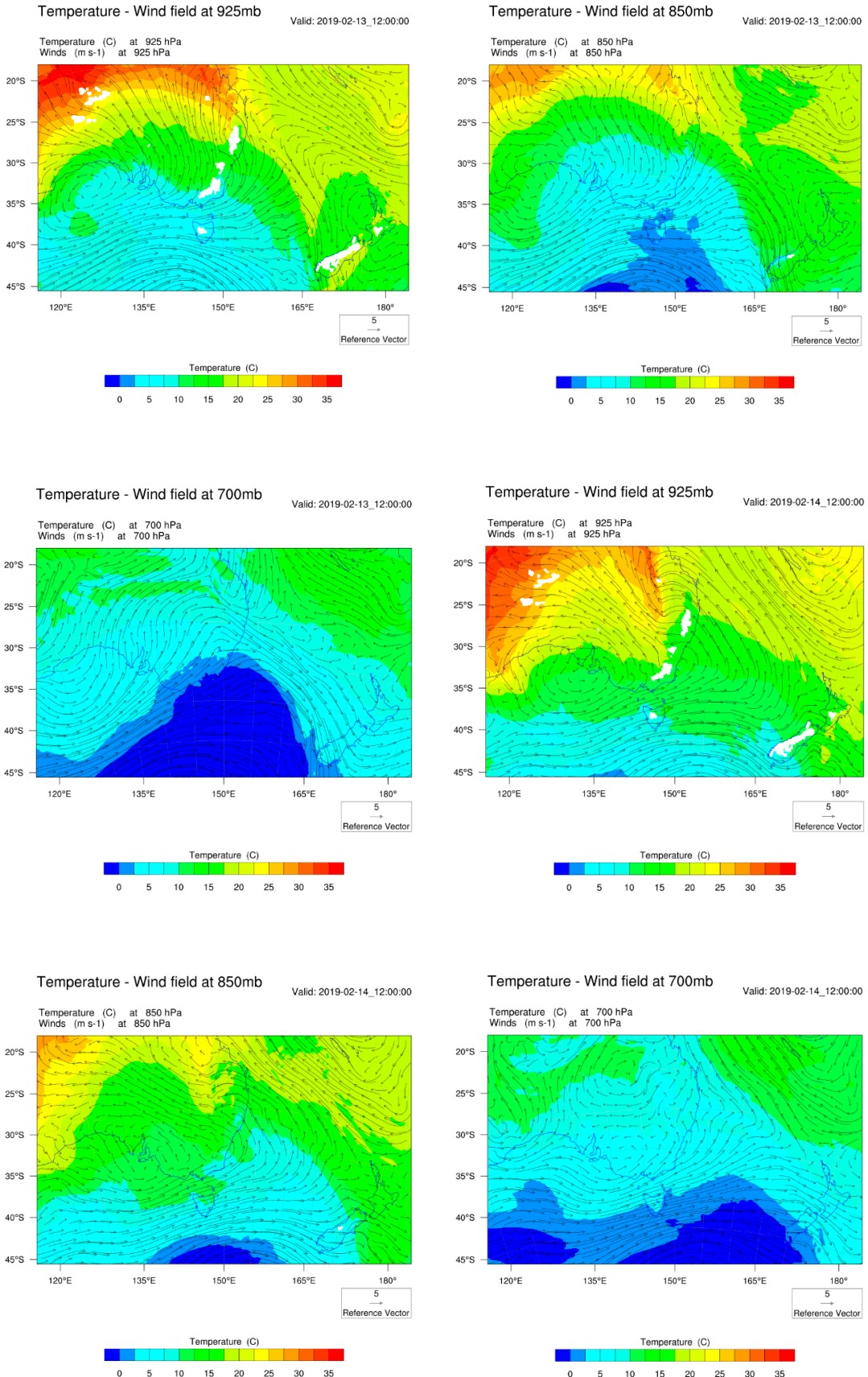

**Figure A2.** Temperature and wind field at 950 mb, 850 mb and 700 mb on 13 February 2019 12:00 UTC and 14 February 2019 12:00 UTC.

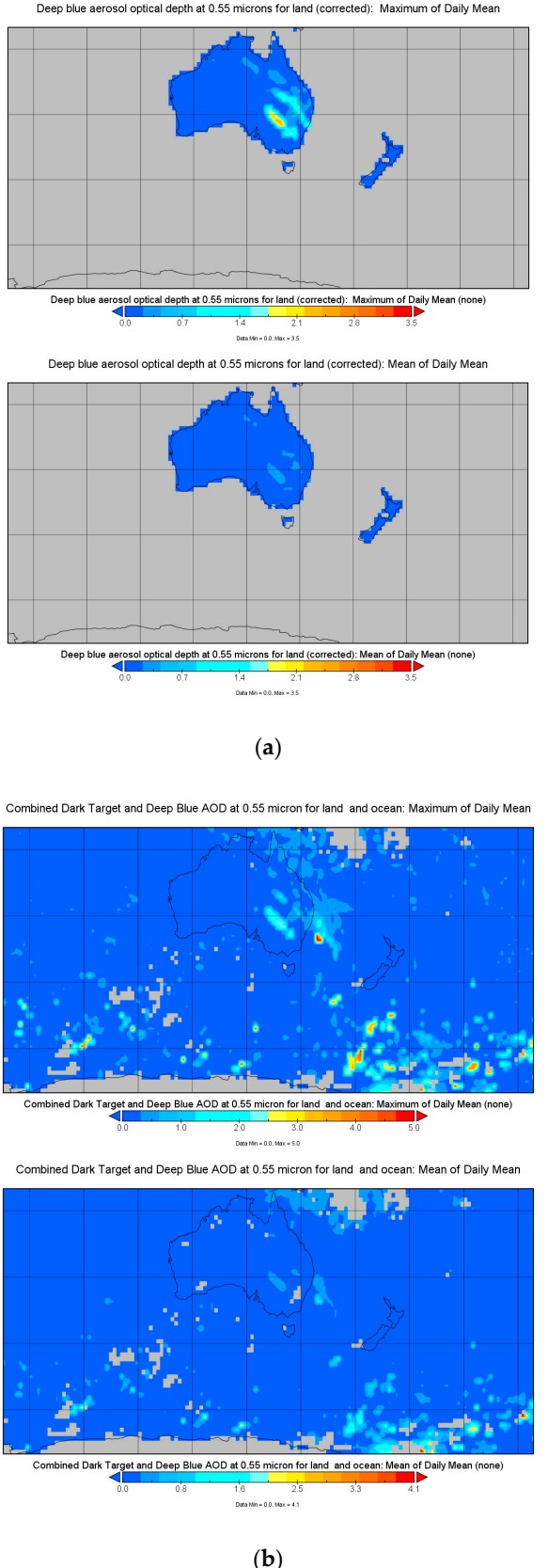

**Figure A3.** (**a**) Aqua MYD08-E3 maximum and mean of daily mean on land for the period from 10 to 13 February 2019 and (**b**) Aqua MYD08-E3 Combined Dark Target (ocean) and Deep Blue (land) maximum and mean of daily mean for the period from 10 to 17 February 2019.

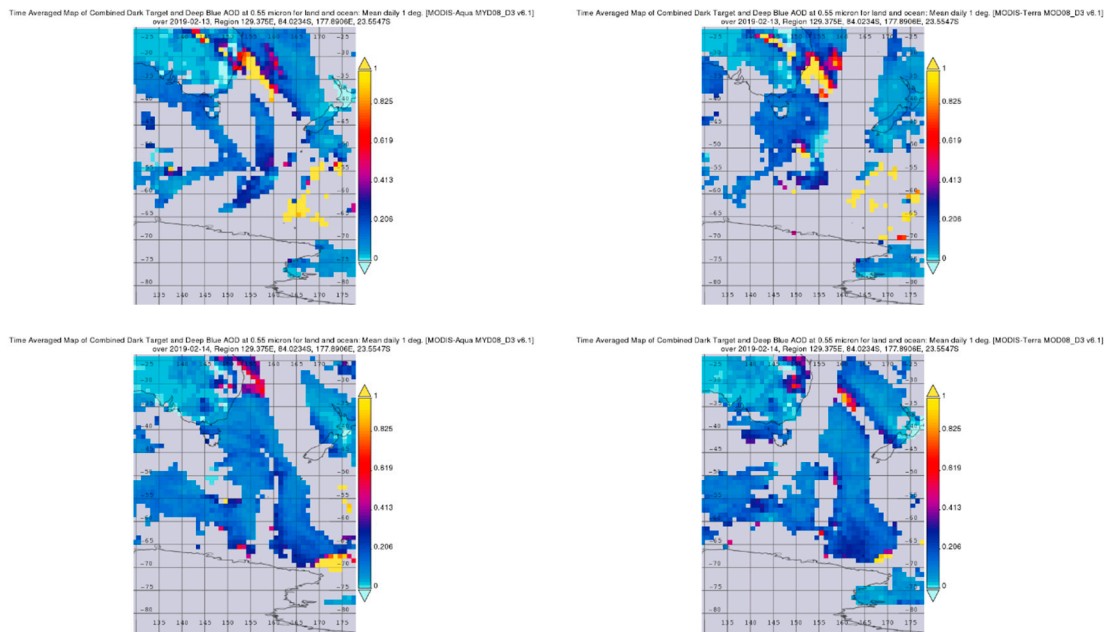

**Figure A4.** Daily average AOD for 13 (top) and 14 February 2019 (bottom) as measured by MODIS Aqua (left) and Terra (right) between Australia, New Zealand and Antarctica (https://giovanni.gsfc.nasa.gov/).

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
