# Peer review of "Dust Storm Event of February 2019 in Central and East Coast of Australia and Evidence of Long-Range Transport to New Zealand and Antarctica"

_atmosphere, doi:10.3390/atmos10110653_

Round 1

Reviewer 1 Report

The topic of this manuscript is interesting. However, this manuscript is not well organized and many content is not related with the topic. I suggest the authors should rewrite the manuscript before reconsideration. And many figures are unclear, and the authors should redraw them.

Specific comments:

Abstract is too long, should be shorten and keep important part.

Line 31-33: ‘This process has been going on 31 for at least the last 170k years …’  There is no evidence to support that the dust in ice cores and sediment records are from this process and this sentence should be delete.

Line 47: ‘the third largest’ is correct but quite misleading, since its emission amount is quite small. The authors should write the fraction of the Australian dust emission amount to the global one.

Figure 2: Some characters are too small to see.

Figure 3: Merge a and b into one figure.

Line 336: Other emission data.

Line 338: The global 2005 anthropogenic emission is too old for simulation the case in 2019. And what’s the horizontal resolution of it?

Figure 5: The contour line makes the figure quite unclear, please redraw this figure.

Figure 6 & 7: Some characters are too small to see. And is so many figures really necessary?

Figure 8: Are all four figures used in the main text? If yes, the authors should write in main text; if not, delete the unnecessary figures. And please do it for other figures. There are too many figures in the manuscript and looks not used in the main text, it’s waste of time for the readers.

Figure 13: Show figure 13a in figure 3. The figure 13b-f is quite unclear, please redraw it. And which is observation, and which is the simulation, write it clearly in the figures.

Author Response

The topic of this manuscript is interesting. However, this manuscript is not well organized and many content is not related with the topic. I suggest the authors should rewrite the manuscript before reconsideration. And many figures are unclear, and the authors should redraw them.

Reply: The authors would like to thank the reviewer for informative and constructive suggestion. Many figures are redrawn and some figures are removed or relegated to the Appendix so the content is consistent and ideas organised to be followed easily.

Specific comments:

Abstract is too long, should be shorten and keep important part.

Reply: The abstract is now shortened

Line 31-33: ‘This process has been going on 31 for at least the last 170k years …’  There is no evidence to support that the dust in ice cores and sediment records are from this process and this sentence should be delete.

Reply: This sentence is now deleted. The sentence referred to studies that identified the dust in ice cores from Dome C as having Pb isotope profile matched with the soil Pb profile from the Murray-Darling basin in Australia 

Line 47: ‘the third largest’ is correct but quite misleading, since its emission amount is quite small. The authors should write the fraction of the Australian dust emission amount to the global one.

Reply: The sentence is now rewritten as 

"Ginoux et al 2004 [6] showed that Australia is the third largest source contributing to global dust emission, after North Africa and Asia. But in term of dust emission amount, it is only about 20% of Asia and 7% of North Africa annual emission amount."

Figure 2: Some characters are too small to see.

Reply: This is now corrected.

Figure 3: Merge a and b into one figure.

Reply: It is difficult to merge the two as the context of the two are different. We decide to retain as it is.

Line 336: Other emission data.

Reply: Listed gases are now shortened as suggested.

Line 338: The global 2005 anthropogenic emission is too old for simulation the case in 2019. And what’s the horizontal resolution of it?

Reply: This is the latest that we can obtain from EDGAR (Emission Database for Global Atmospheric Research) version 4. The resolution is 1 degree by 1 degree. The horizontal resolution is about 110km.

Figure 5: The contour line makes the figure quite unclear, please redraw this figure.

Reply: The figure is now redrawn with contour line spacing is larger.

Figure 6 & 7: Some characters are too small to see. And is so many figures really necessary?

Reply: These figures are now put in the Appendix.

Figure 8: Are all four figures used in the main text? If yes, the authors should write in main text; if not, delete the unnecessary figures. And please do it for other figures. There are too many figures in the manuscript and looks not used in the main text, it’s waste of time for the readers.

Reply: These four figures are used in the main text. We now delete Figure 19, Figure 12 is moved to the Appendix. All Figures are used in the main text in the revised manuscript.   

Figure 13: Show figure 13a in figure 3. The figure 13b-f is quite unclear, please redraw it. And which is observation, and which is the simulation, write it clearly in the figures.

Reply: Figure 13b-f are now redrawn as Figure 11b-f. The blue lines are observation while the red lines are predicted concentration. They are written in the figures.

Reviewer 2 Report

Firstly, the study adds to the understanding of the existing dust transport mechanism in Australia. It extends perspectives by using many scientific tools (e.g. atmospheric modeling, satellite measurement, ground-based observations). I suggest a set of minor updates.

Paper requires a standard in the citation, the paper shall be published only after maintaining uniformity in citations.

There are inconsistencies in tense use. The whole paper needs to be checked and updated.

As far as I see, the authors didn't mention the aim of the study in any section. This needs to be provided to the reader ideally in the instructions.

I also suggest authors mention the meteorological data that they used for their model runs in the “data and methods” section.

Domain selection and the boundary conditions for the WRF-Chem run are well defined. Existing setting provides adequate projections for interpretation and enough. However, considering the total run-time and domain selected, the spatial resolution of the runs could have been improved. The authors shall also mention the limits of their projection/run.

A pdf file including comments/suggestions are also provided to the editor. The editor is welcomed to share this version with the authors if required

Author Response

The authors would like to thank the reviewer for detailed comments on the manuscript which make it better in the revised manuscript.

Citation is now standardised and tenses used are consistent. The aim is specified in the introduction and meteorological data used is specified in the "data and methodology" section.

The detailed suggestions and comments by the reviewers are reflected in the revised manuscript with changes are highlighted (in yellow) for tracking.

Reviewer 3 Report

The authors present an interesting case, during which a dust storm transported dust from the south-east coast of Australia to New Zealand and Antarctica.

They used both model-base tools, i.e. WRF-Chem and HYSPLIT, and satellite observations, i.e. CALIPSO and MODIS AOD data, to support their results.

I think that data are well described and results are sufficiently supported by modeled and experimental data.

No have no specific concern and suggest to publish this work as is.

Best regards

Author Response

The authors present an interesting case, during which a dust storm transported dust from the south-east coast of Australia to New Zealand and Antarctica.

They used both model-base tools, i.e. WRF-Chem and HYSPLIT, and satellite observations, i.e. CALIPSO and MODIS AOD data, to support their results.

I think that data are well described and results are sufficiently supported by modeled and experimental data.

No have no specific concern and suggest to publish this work as is.

Reply: The authors would like to thank the reviewer for the review of our manuscript

Round 2

Reviewer 1 Report

Accept in present form